palaeontology

Cretaceous, Australia, Dinosauria, Theropoda

**Author for correspondence:**
Tom Brougham
e-mail: tbrougha@myune.edu.au

# New theropod (Tetanurae: Avetheropoda) material from the 'mid'-Cretaceous Griman Greek Formation at Lightning Ridge, New South Wales, Australia

Tom Brougham[1], Elizabeth T. Smith[2] and Phil R. Bell[1]

[1]School of Environmental and Rural Science, University of New England, Armidale 2351, New South Wales, Australia
[2]Australian Opal Centre, 3/11 Morilla Street, Lightning Ridge 2834, New South Wales, Australia

TB, 0000-0002-2771-536X; PRB, 0000-0001-5890-8183

The limited fossil record of Australian Cretaceous theropods is dominated by megaraptorids, reported from associated and isolated material from the Early Cretaceous of Victoria and the 'Mid'-Cretaceous of central-north New South Wales and central Queensland. Here, we report on new postcranial theropod material from the early Late Cretaceous Griman Creek Formation at Lightning Ridge. Among this new material is an associated set consisting of two anterior caudal vertebrae and a pubic peduncle of the ilium, to which a morphologically similar partial vertebral centra from a separate locality is tentatively referred. These elements display a combination of characteristics that are present in megaraptorid and carcharodontosaurid theropods, including camellate internal organization of the vertebral centra, ventrally keeled anterior caudal centra and a pubic peduncle of the ilium with a ventral surface approximately twice as long anteroposteriorly as mediolaterally wide. Unfortunately, a lack of unambiguous synapomorphies precludes accurate taxonomic placement; however, avetheropodan affinities are inferred. This new material represents the second instance of a medium-sized theropod from this interval, and only the third known example of associated preservation in an Australian theropod. Additional isolated theropod material is also described, including an avetheropodan femoral head that shows similarities to *Allosaurus* and *Australovenator*, and a mid-caudal vertebral centrum bearing pneumatic foraminae and extensive camellae that is referrable to Megaraptora and represents the first axial skeletal element of a megaraptorid described from Lightning Ridge.

# 1. Introduction

The fossil record of Australian Cretaceous theropods is scarce and is composed almost exclusively of isolated and fragmentary remains [1]. The majority of the reported Australian theropod skeletal material to date has come from the diverse high-latitude fauna of the Aptian–Albian Otway and Gippsland groups of southern Victoria, consisting of isolated individual elements of megaraptorans, maniraptorans, ceratosaurians, spinosaurids and putative tyrannosauroids [1–8]. By contrast, the most complete Australian theropod is the megaraptorid *Australovenator wintonensis* from the Cenomanian–Turonian Winton Formation of central Queensland, known from mandibular, forelimb, hindlimb and pelvic elements [9–12]. Despite the evidence for a high diversity of theropods in Australia, the record of apex theropod predators appears to be dominated by megaraptorids. This is in contrast to the contemporaneous theropod fossil record of Patagonia, at roughly the same palaeolatitude, which hosted a diverse range of abelisaurids, with a smaller component of the fauna consisting of carcharodontosaurids and megaraptorids [13].

The Griman Creek Formation (GCF) at Lightning Ridge in northern New South Wales preserves one of the most diverse Australian Cretaceous terrestrial faunal assemblages [14,15], the vertebrate component of which has received little attention until recently. The first named Australian theropod, *Rapator ornitholestoides*, was described on the basis of a single metacarpal I discovered in the GCF in the vicinity of Lightning Ridge [16]. While this taxon is now considered to be a *nomen dubium*, subsequent comparisons with the same element in *Australovenator* and *Megaraptor* indicate megaraptorid affinities for *Rapator* [1,12,17]. More recently, the associated remains of a megaraptorid were described from a proximal ulna, proximal manual ungual, pubic peduncle of the ilium, fibula and metatarsal III [18]. Aside from *Australovenator*, this specimen represents only the second example of associated preservation of theropod remains in Australia. Here, we report on new postcranial theropod material, including an association of caudal vertebrae and pelvic elements, from the GCF near Lightning Ridge, New South Wales. We also describe an isolated femoral head of an indeterminate avetheropodan and a mid-caudal vertebral centrum that represents the first axial vertebral element of a megaraptorid to be described from Lightning Ridge.

# 2. Institutional abbreviations

AM (Australian Museum, Sydney, New South Wales); LRF (Australian Opal Centre, Lightning Ridge, New South Wales).

# 3. Locality and geological setting

All fossils were excavated from subsurface beds of the Griman Creek Formation as a result of opal mining activity in the vicinity of Lightning Ridge, central-northern New South Wales, Australia (figure 1). The Griman Creek Formation is situated within the Surat Basin, which extends over southeastern Queensland and northern New South Wales. The Eromanga Basin neighbours the Surat Basin to the west (figure 1). Together, these two basins mark the maximum transgression of the Eromanga Sea, which persisted across the central part of Australia for much of the Early Cretaceous and up to the Cenomanian, and form the majority of the present-day Great Artesian Basin. The Griman Creek Formation is composed of thinly laminated and interbedded fine- to medium-grained sandstones, siltstones and mudstones, with carbonate cements, intraformational conglomerate beds and coal deposits [20,21]. Within the Griman Creek Formation, opal and fossils occur within interbedded siltstone and mudstone layers, often referred to informally as the 'Finch clay' facies [22]. Preservation of fossils at Lightning Ridge—including those specimens described here—is commonly in the form of natural casts, or pseudomorphs, in non-precious opal [18,23,24]. The depositional environment of the Griman Creek Formation is interpreted as a lacustrine to estuarine coastal floodplain with fluvial and deltaic influences [18]. New radiometric dates (Bell *et al.* [25]) for the Griman Creek Formation indicate an early Cenomanian (100.2–96.6 Ma) maximum depositional age is significantly younger than Late Albian age, which had previously been assigned on the basis of palynomorphs [22] and radiometric dates from fission track analysis of detrital zircons obtained from subsurface samples of the Griman Creek Formation in Queensland [26]. Consequently, deposition of the fossiliferous part of Griman Creek Formation at Lightning Ridge took place during the early Cenomanian,

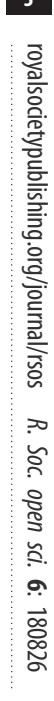

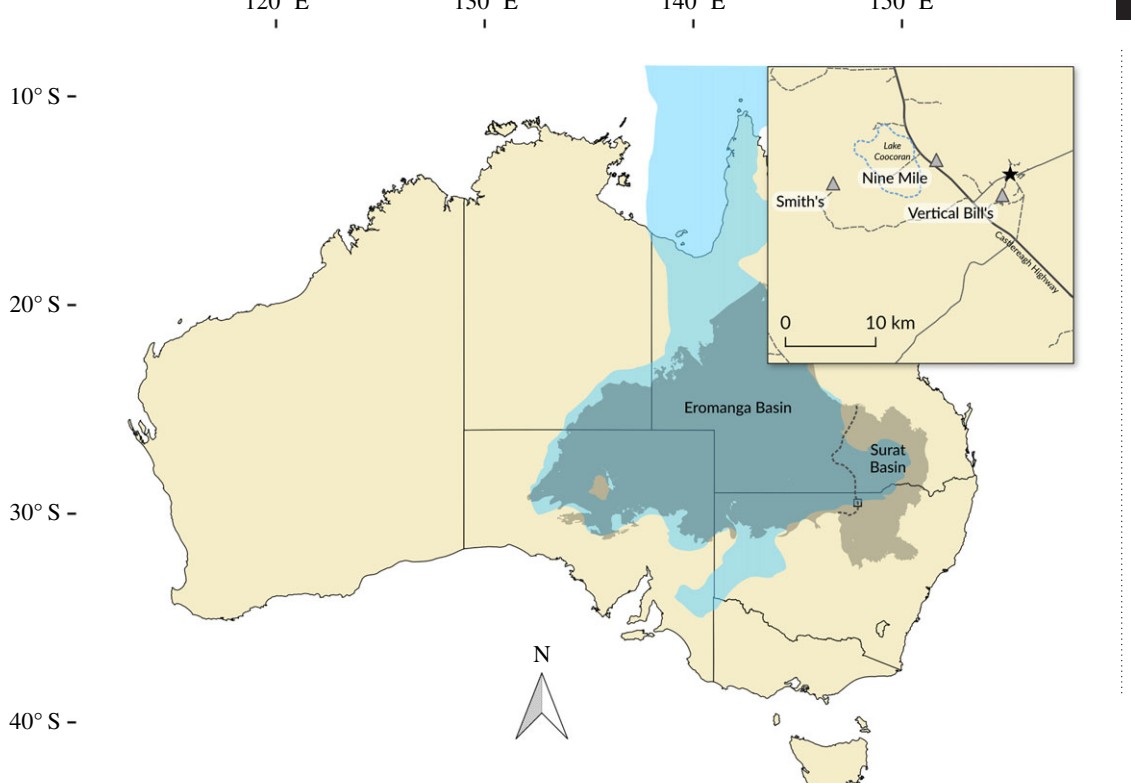

**Figure 1.** Map of Australia showing the location of Lightning Ridge and the mineral claims in which the fossils were recovered. The extent of Cretaceous Eromanga and Surat basins is represented by the grey area separated by dashed line. The extent of the Eromanga and Surat Basins is represented by the grey area separated by the dashed line. The maximum extent of the Eromanga Sea is indicated by the area in blue. Australia coastline uses data taken from GEODATA COAST 100 K 2004 provided by Geoscience Australia (http://www.ga.gov.au/metadatagateway/metadata/record/61395). Basin extents use data taken from [19].

penecontemporaneous with the deposition of the fossiliferous upper portion of the Winton Formation [25,27].

The Griman Creek Formation at Lightning Ridge preserves a rich array of vertebrate and invertebrate fauna, including crocodylomorphs [23,28,29], australosphenidian mammals [30–32], ornithischian dinosaurs [33–35], titanosauriform sauropods [36], megaraptoran theropods [18,37], enantiornithine birds [38], pterosaurs [39], plesiosaurs [40], turtles [41,42], dipnoan lungfish [43–45], a possible synapsid [24] and numerous species of non-marine macro-invertebrates [22,46–48]. For a review of the vertebrate fauna of the Griman Creek Formation, see Bell *et al*. [25].

## 4. Systematic framework

The phylogenetic position of Megaraptora has been the subject of much recent debate. Megaraptora was originally defined as a clade consisting of *Megaraptor*, *Australovenator*, *Orkoraptor*, *Fukuiraptor*, *Aerosteon* and *Chilantaisaurus*, as the sister taxon to *Neovenator* to form the clade Neovenatoridae. Neovenatoridae and its sister taxon Carcharodontosauridae together formed Carcharodontosauria, which in turn was the sister taxon to *Allosaurus* [49]. A review of the fossil record of Patagonian theropods identified a number of morphological similarities between megaraptorans and tyrannosauroids [13], and in a novel phylogenetic analysis that incorporated characters and taxa from existing datasets focused separately on basal tetanurans and tyrannosauroids [49,50] hypothesized a deeply nested position of Megaraptora, with the exclusion of *Chilantaisaurus*, within Tyrannosauroidea. The clade Megaraptoridae was also erected to encompass all Gondwanan megaraptorans. This hypothesis was maintained in a phylogenetic analysis that included new observations of *Megaraptor* based on a partially complete juvenile specimen [51]. However, a subsequent phylogenetic analysis accompanying the description of the bizarre tetanuran *Gualicho*, based on a modified version of the Porfiri *et al*. dataset, hypothesized

that Megaraptora was the sister taxon of Coelurosauria [52]. In addition, a review of megaraptorid manual anatomy concluded that, while sharing some derived characters with coelurosaurs, and tyrannosauroids less inclusively, *Australovenator* and *Megaraptor* retained most of the plesiomorphic characters of the manus as present in *Allosaurus*, thus constituting evidence against the hypothesis of tyrannosauroid affinities for megaraptorans [53]. Clearly, the phylogenetic position of Megaraptora remains an unresolved problem; however, the composition of Megaraptora and Megaraptoridae has remained remarkably consistent among the different hypotheses. Accordingly, in the following discussion we refrain from preferring any of the three prevailing hypotheses but follow the taxonomic composition of Megaraptora and Megaraptoridae as defined by Novas *et al.* [13].

Nomenclature for description of vertebral laminae and fossae follows that of [54] and [55], respectively.

# 5. Systematic palaeontology

Dinosauria Owen, 1842
Theropoda Marsh, 1881
Tetanurae Gauthier, 1986
Avetheropoda Paul, 1988
Avetheropoda indet.

## 5.1. LRF 3310–3312, AM F106525

### Material

Two anterior (LRF 3310, LRF 3311), a right pubic peduncle of the ilium (LRF 3312) and a centrum articular surface (AM F106525). LRF 3310–3312 were recovered from a one-metre diameter drill shaft in the eastern section of Smiths Field on the Coocoran opal field, approximately 20 km west of Lightning Ridge (figure 1). Their close association within the thin opal- and fossil-bearing layer of the GCF and the absence of overlapping material or other taxa in the immediate vicinity indicates that they pertain to a single individual. AM F106525 was recovered from a mineral claim known as The Bone Yard at the Nine Mile field, approximately 8 km west-northwest of Lightning Ridge.

### Preservation

#### LRF 3310–3312

LRF 3310 represents an almost complete centrum, the posterior part of the neural arch, and the base of the right transverse process (figure 2). The anterior end of the vertebral centrum has been abraded but is intact (figure 2*b*). The posterior articular surface of the centrum is well preserved but is missing a portion of the left rim (figure 2*a*). The left lateral surface of the centrum has been crushed (figure 2*d*), resulting in a rightward displacement of the anterior end of the centrum in ventral view (figure 2*e*). Diagenetic veins of silica have formed in and around the crushed area on the left side of the centrum; this mineralization can also be seen on the neural spine and postzygapophysis (figures 2*d* and 3). Of the neural arch, only the right postzygapophysis and the bases of the right transverse process and neural spine are preserved. The edges of the articular surface of the postzygapophysis have been eroded (figure 2*c*).

LRF 3311 represents the ventral portion of a centrum and a dorsal fragment of the posterior articular surface (figure 4). The anterior articular end of the centrum has been broken off, exposing an internal cavity (figure 4*b*); no internal structures can be discerned along the plane of the break.

LRF 3312 is interpreted as representing the ventral end of the pubic peduncle of a right ilium. The broken and exposed dorsal surface is mediolaterally thin, and the interior of the bone appears to have been preserved as a solid mass of opal, obscuring any detail of the original bone texture. The lateral surface is well preserved whereas only the dorsalmost portion of the medial surface of the peduncle is visible through the adherent matrix (figure 5*a,b*). The ventral surface, where it would have contacted the proximal pubis, is heavily eroded and densely covered in matrix. On the concave posteroventral surface, two subcircular depressions are present (figure 5*d*) that are inferred to be possible bioerosional features, and as such do not represent an original feature of the bone. Only the ventrolateral portion of the acetabular margin is preserved.

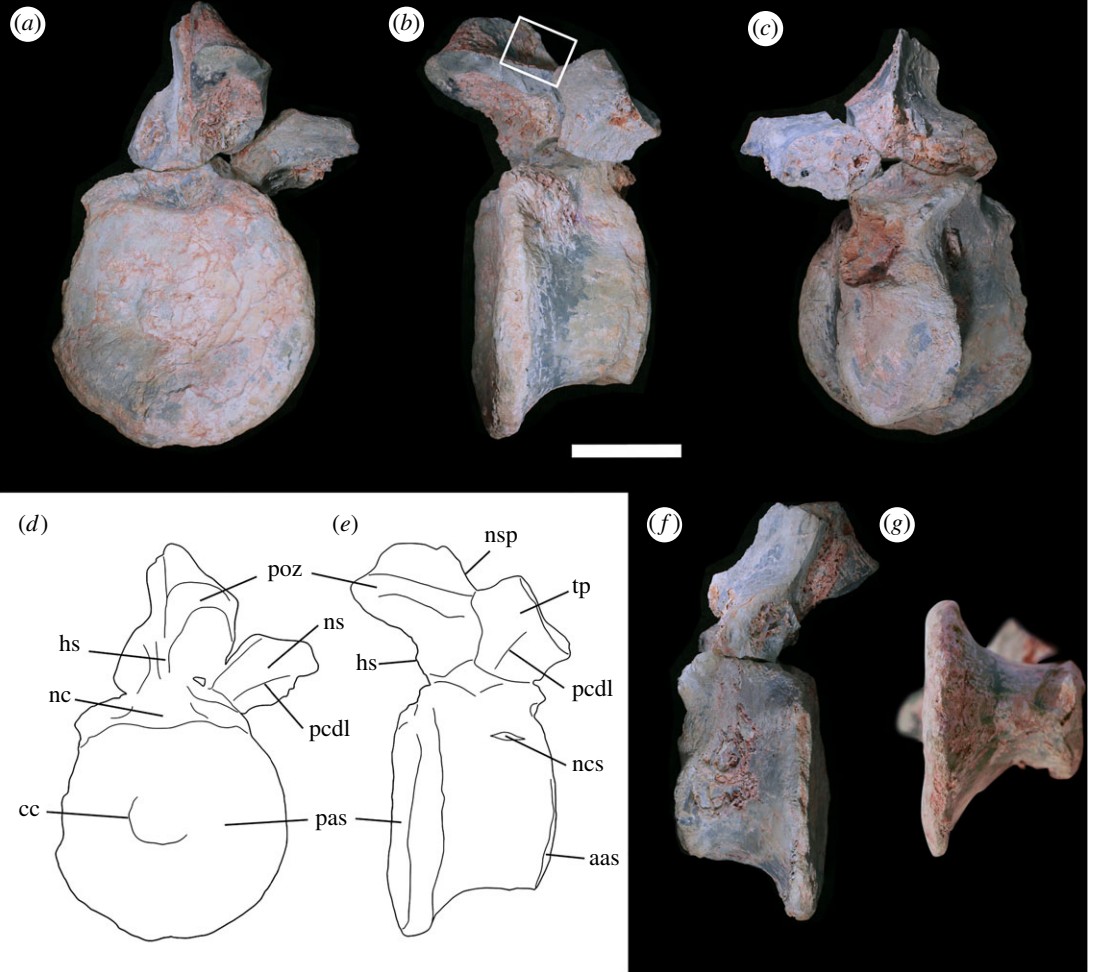

**Figure 2.** Anterior caudal vertebra LRF 3310 in (*a,d*) posterior, (*b,e*) right lateral, (*c*) anterior (*f*) left lateral (*g*) ventral views. Boxed area on (*b*) is expanded in figure 3. aas, anterior articular surface; cc, central convexity; hs, hyposphene; nc, neural canal; ncs, neurocentral suture; nsp, neural spine; pas, posterior articular surface; pcdl, posterior centrodiapophyseal lamina; poz, postzygapophysis. Scale bar equals 50 mm.

## AM F106525

AM F106525 is an isolated articular end of a centrum (figure 6); only a thin portion of the centrum is preserved.

## Description

### LRF 3310–3312

As preserved, the centrum of LRF 3310 is markedly shorter anteroposteriorly than the dorsoventral height of the posterior articular surface (table 1). The posterior articular surface is subcircular and slightly concave, the degree of concavity stronger towards the centre of the articular surface. A small subcircular convexity is present in the centre of the posterior articular surface, approximately one-third (31%) of the width and height of the articular surface itself (figure 2*a,e*). The rim of the posterior articular surface is thickened, the dorsal margin of which is depressed in posterior view, forming a trough level with the floor of the neural canal. There is no indication of a chevron facet on the posteroventral part of the centrum (figure 2*b,g*). The anterior articular surface is a dorsoventrally elongate ellipse in anterior view. The dimensions of the anterior articular surface are markedly less than those of the posterior articular surface. In lateral view, the ventralmost extent of the anterior articular surface is dorsally offset relative to that of the posterior articular surface (figure 2*b,c*). The lateral surfaces of the centrum are smooth, concave anteroposteriorly and convex dorsoventrally, meeting ventrally to form a gently curved surface with no sign of either a ventral keel or midline

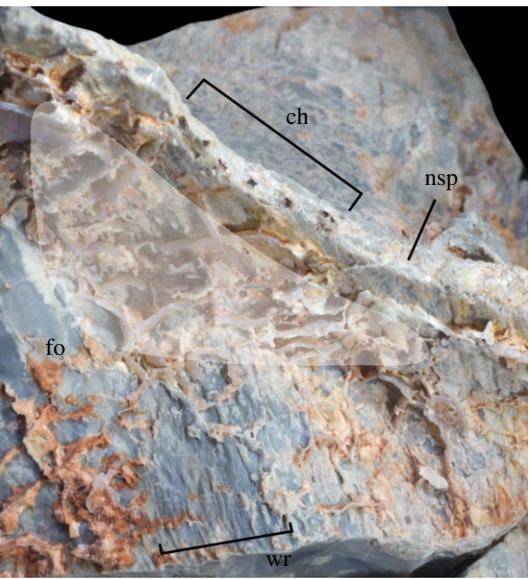

**Figure 3.** The right lateral surface of the neural arch of the anterior caudal vertebra LRF 3310, showing the location of the postzygapophyseal spinodiapophyseal fossa (grey). ch, silica-filled channels; fo, postzygapophyseal spinodiapophyseal fossa; nsp, neural spine; wr, wrinkled texture.

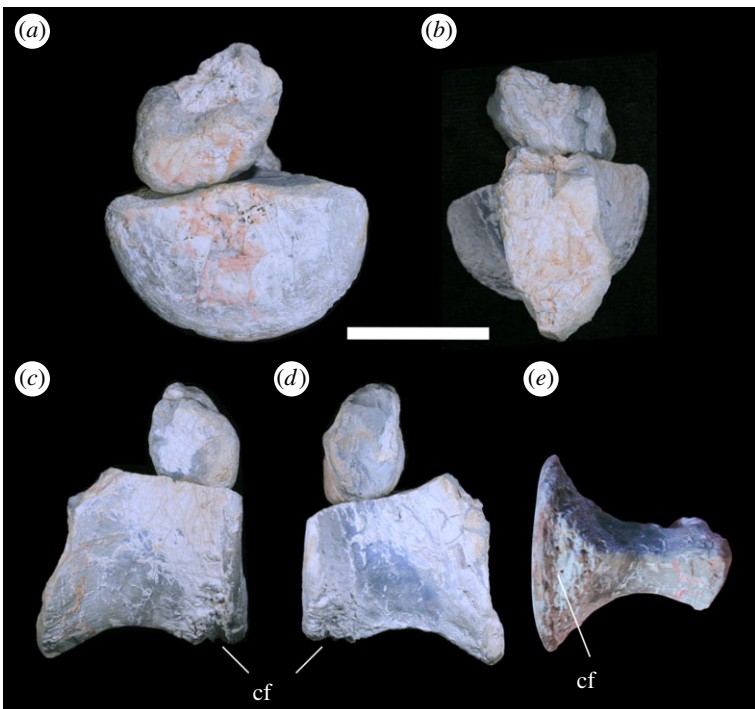

**Figure 4.** Anterior caudal vertebra LRF 3311 in (*a*) posterior, (*b*) anterior, (*c*,*d*) lateral and (*e*) ventral views. cf, chevron articular facet. Scale bar equals 50 mm.

groove. No pneumatic foramina are visible on the preserved lateral surfaces of the centrum. On the right lateral surface, a slight depression is present at the point of contact between the centrum and the neural arch, associated with traces of the closed neurocentral suture (figure 2*b*). An oblique fracture at the anterodorsal surface of the centrum reveals polygonal regions bounded by thin septa that represent the camellate internal structure of the centrum (figure 2*d*).

The base of the transverse process of LRF 3310 is dorsoventrally compressed and extends laterally and horizontally from the neural arch. A weakly developed posterior centrodiapophyseal lamina is present on the posteroventral surface of the transverse process. A robust postzygadiapophyseal lamina extends between the transverse process and postzygapophysis but is partially broken. The postzygapophysis is

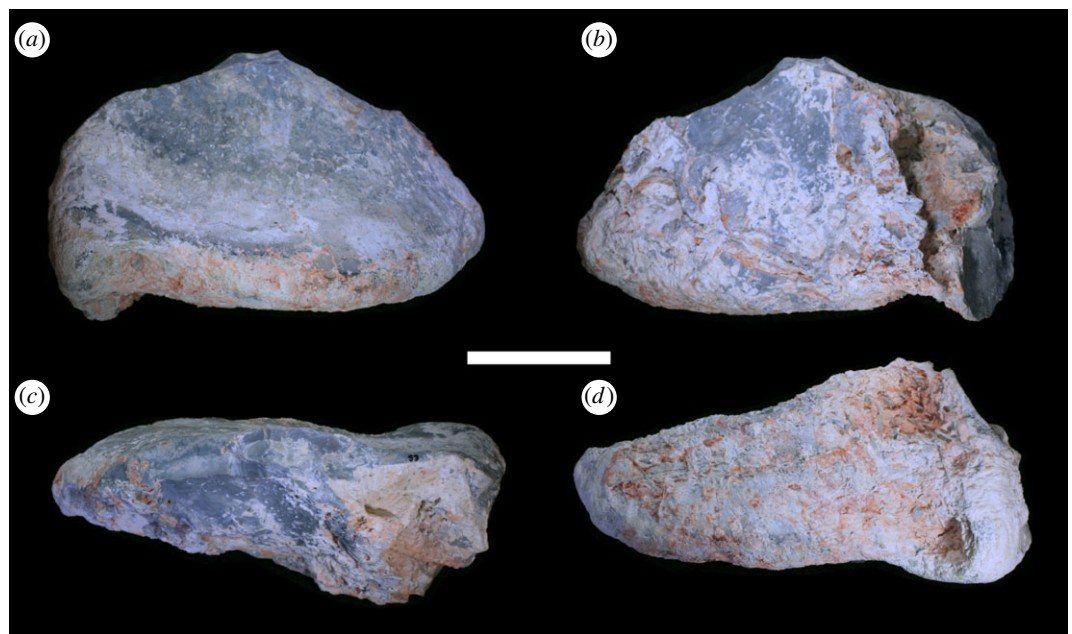

**Figure 5.** Right pubic peduncle of the ilium LRF 3312 in (*a*) lateral, (*b*) medial, (*c*) dorsal and (*d*) ventral. Scale bar equals 50 mm.

**Table 1.** Selected measurements for the vertebrae LRF 3310, LRF 3311 and AM F112816. All measurements in millimetres. Asterisks indicate that measurements are from the posterior end of the centrum.

| measurement | LRF 3310 | LRF 3311 | AM F112816 |
|---|---|---|---|
| centrum, anterior articular surface, width | 47.3 | — | — |
| centrum, anterior articular surface, height | 58.5 | — | 7.0 |
| centrum, posterior articular surface, width | 101.6 | 73.4 | 26.3 |
| centrum, posterior articular surface, height | 93.3 | — | 32.4 |
| centrum, anteroposterior length | 58.3 | 65.6 | 44.9 |
| centrum, mediolateral width at mid-length | 41.5 | 26.7 | 17.1 |
| centrum, dorsocentral height at mid-length | 58.1 | — | 32.9 |
| neural arch, height | 20.3 | — | — |
| neural canal, width | 30.8* | — | — |
| neural canal, height | 20.3* | — | — |

robust and extends beyond the posterior articular surface of the centrum (figure 2*b,c*). The articular surface of the postzygapophysis faces mostly ventrally with a slight posteromedial inclination; it is subrectangular and slightly longer anteroposteriorly than wide mediolaterally. A hyposphene is present immediately ventromedially to the postzygapophysis. The base of the neural spine is positioned towards the posterior end of the centrum and is laterally compressed.

A depression is present at the base of the neural spine of LRF 3310 on the right side, anterior to the postzygapophysis; this surface is not preserved on the left side. This is interpreted here as representing a postzygapophyseal spinodiapophyseal fossa (posdf; figure 3). The dorsal surface of the neural arch in the vicinity of this fossa bears a wrinkled texture that extends almost perpendicular to the neural spine and continues along the transverse process parallel to its posterior edge (figure 3). The wrinkled texture also appears to extend onto the lateral surface of the neural spine; however, diagenetic veins of silica within the fossa obscure much of this surface. The wrinkled texture does not extend posteriorly onto the base of the postzygapophysis. The posdf is most deeply impressed at its posteriormost extent, immediately anterior to the postzygapophysis. On the broken cross-section of the neural spine and dorsal to the posdf, six equally spaced black marks are visible. These marks appear to represent the infilling of channels or foramina within the neural spine by silicate minerals; the presence of diagenetic silica

veins on LRF 3310 appears to coincide with breakages or localized erosion on the external surface of the bone (see Preservation section above) that exposed internal structures within the centrum or neural arch. However, the posdf appears to have sustained no significant erosion or fractures as seen on other areas where diagenetic silica veins are observed. Therefore, based on the presence of diagenetic silica both within the posdf and channels within the neural spine, it is suggested that the posdf may have borne one or more foramina and thus served a pneumatic function.

The posterior articular surface of LRF 3311 is subcircular and slightly concave with a well-developed rim (figure 4a). Immediately anterior to this rim and on the ventral surface of the centrum is a transverse groove which probably represents an articular facet for the chevron (figure 4c,d). While the anterior articular surface is missing, comparisons with LRF 3310 indicates that the anterior articular surface is likely to have been smaller than the posterior articular surface. In ventral view, the centrum is strongly constricted; the width of the centrum at its narrowest point is approximately one-third of the mediolateral width of the posterior articular surface (figure 4e and table 1). The lateral surfaces of the centrum are concave anteroposteriorly, gently convex dorsoventrally and, unlike in LRF 3310, converge ventrally to form a well-defined keel. A series of faint longitudinal striations are visible on the posterolateral surfaces of the centrum (figure 4e). The ventral margin of the centrum is concave in lateral view (figure 4c,d). Anteriorly, the ventralmost preserved point of the centrum extends ventral to the ventralmost point of the posterior articular surface, indicating that the missing anterior articular surface was ventrally offset relative to the posterior articular surface. Dorsally, the ventral outline of the neural canal is visible.

The pubic peduncle of the right ilium (LRF 3312) is roughly trapezoidal in lateral view (figure 5a,b). The anterior margin is oriented approximately $60°$ relative to the ventral margin. The posterior (acetabular) margin is subvertical and mediolaterally convex ventrally (in posterior view); however, due to erosion, the shape of the rest of the acetabular margin cannot be determined. In lateral view, the ventral margin forms a sinuous contour, the anterior two-thirds being convex whereas the posterior third is concave. In ventral view, the articular surface forms an isosceles triangle; its mediolaterally widest point forming the posterior end and tapering to its narrowest point anteriorly. The ventral surface is approximately twice as long anteroposteriorly as it is mediolaterally wide. The medial and lateral faces are weakly concave dorsoventrally and divergent towards the ventral surface. The anterior third of the lateral surface is ornamented by a series of parallel striae that extend parallel to the anterior margin of the peduncle in lateral view and are roughly equally spaced, approximately 1.5 mm apart. The striae are most densely packed ventrally (figure 5a). There is no evidence of pneumaticity either in the form of internal cavities on the broken dorsal surface or foramina on the medial or lateral surfaces.

## AM F106525

The articular surface of the centrum is subcircular; its dorsoventral height was probably close to that of the mediolateral width when complete, and therefore similar in shape to the articular surfaces of LRF 3310 and LRF 3311 (figure 6a). The articular surface is slightly concave mediolaterally and dorsoventrally. A small subcircular convexity is present in the centre of the articular surface, similar to that seen on LRF 3310; this feature occupies approximately 27% of the mediolateral width of the articular surface (figures 2a and 6). The small ventral portion of the body of the centrum is concave anteroposteriorly, narrowing to approximately half the mediolateral width of the articular surface (figure 6c) and is slightly concave dorsoventrally. The broken lateral edges of the centrum indicate that the centrum became progressively wider dorsally (figure 6b,f). Ventrally, the surface of the centrum is flattened with no indication of a midline groove or keel (figure 6d,e).

The broken surface of the centrum exposes internal cavities—two of which can be readily distinguished (figure 6b,g)—that extend inside the centrum towards the articular surface. As preserved, the more medial of these cavities extends further into the centrum than the lateral cavity; the septa of the former cavity converge to form an acute angle both immediately behind the articular surface and on the ventral floor of the cavity (figure 6b,f). Another partially preserved septa extends across the dorsal and ventral portions and overlies the cavity, and probably forms the interior boundaries of this cavity (figure 6g); thus, the cavities appear to be dorsoventrally elongate but anteroposteriorly narrow. The cavities are delimited mediolaterally by septa, approximately 1–2 mm thick, that radiate dorsolaterally from the ventromedial margin of the centrum, such that the cavities become broader towards the dorsal surface of the centrum. These cavities are interpreted to form part of a system of internal camerae that extend along the length of the centrum.

Avetheropoda indet.

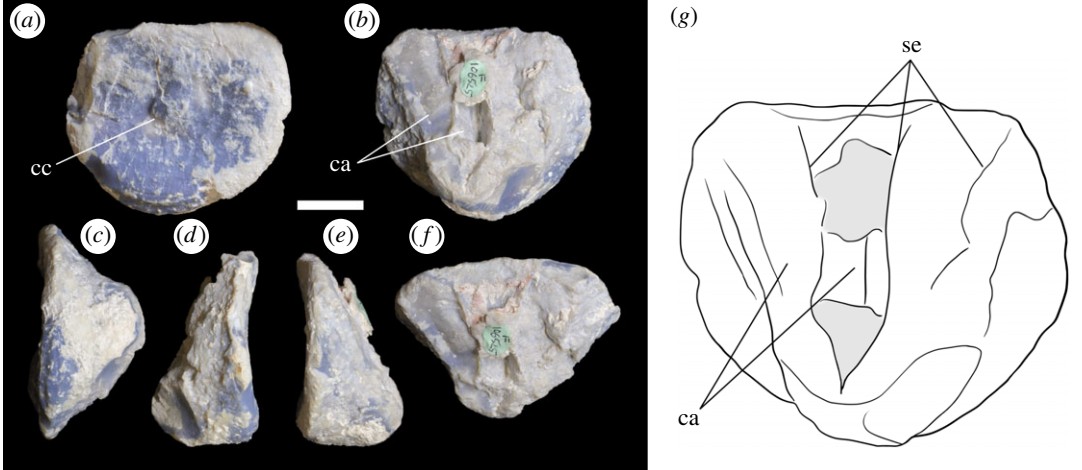

**Figure 6.** Vertebral articular end AM F106525. (*a*) Articular surface of the centrum; (*b*) view of broken surface of the centrum; (*c*) ventral surface; (*d,e*) lateral surfaces; (*f*) oblique dorsal view; (*g*) interpretive drawing of the exposed internal structure of the centrum, grey indicates presence of interior septa overlying the camerae. cc, central convexity; ca, camerae; se, septa. Scale bar equals 20 mm.

## 5.2. AM F105662

### Material

Proximal end of a right femur, including the femoral head (caput) and most of the greater trochanter. The exact location at which this specimen was collected is presently unknown.

### Preservation

The femur below the distal extent of the metaphysis has been lost. The preservation of the femoral head as a pseudomorph has resulted in the loss of some of the osteological correlates for muscular and cartilaginous attachments typically observed in well-preserved theropod femora.

### Description

The femoral head forms an elliptical hemisphere, proximodistally taller than it is anteroposteriorly wide (figure 7*a*–*c*). As the portion of the femur distal to the femoral head is missing, it is not possible to determine the inclination of the head in either the anterior–posterior or dorsoventral planes. However, based on a comparison with the femoral head of *Allosaurus* [56] it is most probable that the femoral head was oriented such that the proximal end of the femur was essentially horizontal in anteroposterior view. In proximal view, the femoral head narrows slightly anteroposteriorly towards the trochanteric end (figure 7*b*), at which point the proximal surface curves down onto the posterior surface (figure 7*d*). The proximal surface of the femoral head, extending laterally towards the greater trochanter, is straight in anterior–posterior views, and the surface texture appears to be smooth, with no noticeable rugosities or indication of a cartilage cone trough (figure 7*b*). A deep groove is present on the posterior surface of the femoral head extending ventrolaterally immediately lateral to the articular surface (figure 7*d*); this denotes the passage of the ligamentum capitis femoris. The fovea capitis is present at the dorsal margin of the groove for the ligamentum capitis femoris and is convex as in most theropods in which it is preserved [57]. However, in AM F105662 its surface is excavated by two narrow bifurcating grooves that diverge from the proximal extent of the groove for the ligamentum capitis femoris (figure 7*b*–*d*). The two grooves diverge at an angle of approximately 50°. Anteriorly, the metaphyseal collar extends from the femoral growth plate to the distalmost preserved extent of the femur proximally. The metaphyseal collar is ornamented by striae (transphyseal striations) that extend roughly perpendicular to the capital-trochanteric axis (figure 7*a*) and represent the extent of the fibrocartilage sleeve on the metaphysis.

Megaraptora [13,49]
Megaraptora indet.

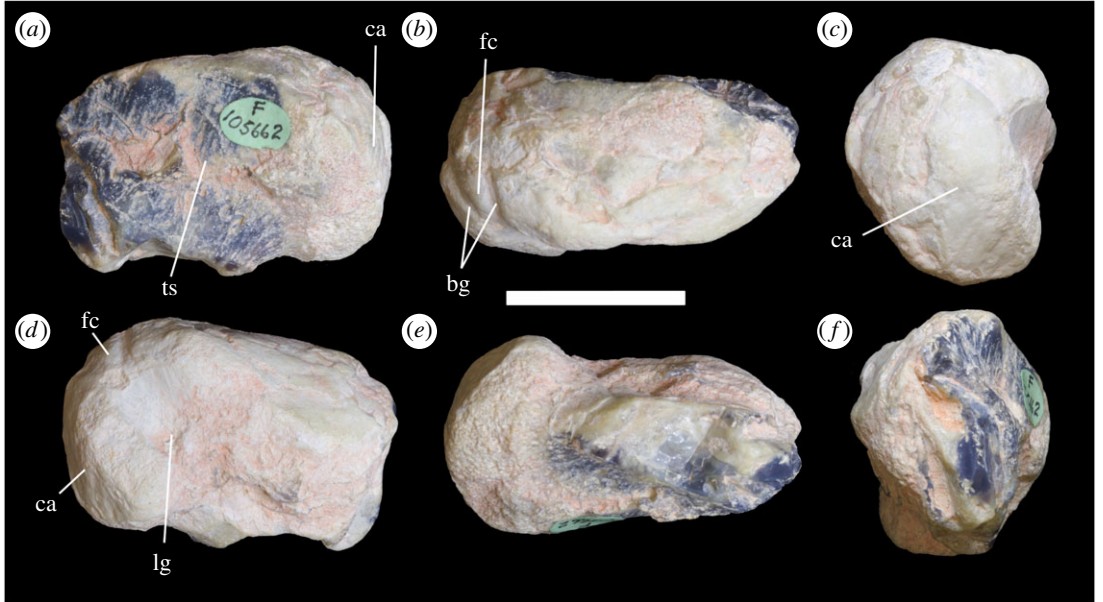

**Figure 7.** Right femoral head AM F105662 in (*a*) anterior, (*b*) proximal, (*c*) medial, (*d*) posterior, (*e*) distal and (*f*) lateral views. bg, bifurcating grooves; ca, caput; fc, fovea capitis; lg, groove for ligamentum capitis femoris; ts, transphyseal striations. Scale bar equals 30 mm.

## 5.3. AM F112816

### Material

AM F112816 is a mid-caudal vertebral centrum and was recovered from Vertical Bill's near Three Mile, approximately 5 km south-southwest of Lightning Ridge (figure 1).

### Preservation

The articular ends and the bases of the exposed neural arch pedicels have been eroded, the anterior more strongly than the posterior; the neural arch is missing.

### Description

Both articular surfaces of AM F112816 are subtriangular, reaching their widest point dorsally and tapering to a blunt point ventrally (figure 8*b,e*). The articular surfaces are flat to slightly concave. In the lateral view, the ventral surface of the centrum is slightly concave and the anterior articular surface is slightly elevated dorsally with respect to the posterior surface (figure 8*a,d*); however, this is probably a result of erosional loss of the ventral rim of the articular surface. The posteroventral surface of the centrum is bevelled to form a chevron articular facet, indicating that it comes from the caudal series. The lateral surfaces of the centrum are slightly concave anteroposteriorly and convex dorsoventrally, the lateral surfaces of the centrum converging ventrally to form a well-defined keel (figure 8*f*). Erosion of the articular and dorsolateral surfaces of the centrum exposes large areas of well-defined polygonal cavities delimited by thin septa within the centrum, representative of camellate pneumaticity. The right surface of the centrum is pierced by two anterior and posterior pneumatic foramen (figure 8*a*). The posterior foramina is subdivided by a thin anterodorsally oriented lamina. There are no pneumatic foramina visible on the left side of the centrum.

## 6. Discussion

LRF 3310–3312 is identified as a theropod primarily due to the presence of hyposphene–hypantrum articulations in the anterior caudal vertebrae, dorsoventrally compressed transverse processes elevated above the dorsal margin of the vertebral centra, weakly developed vertebral laminae and fossae, and camellate internal composition of the vertebral centra. The only Australian non-theropod dinosaurs

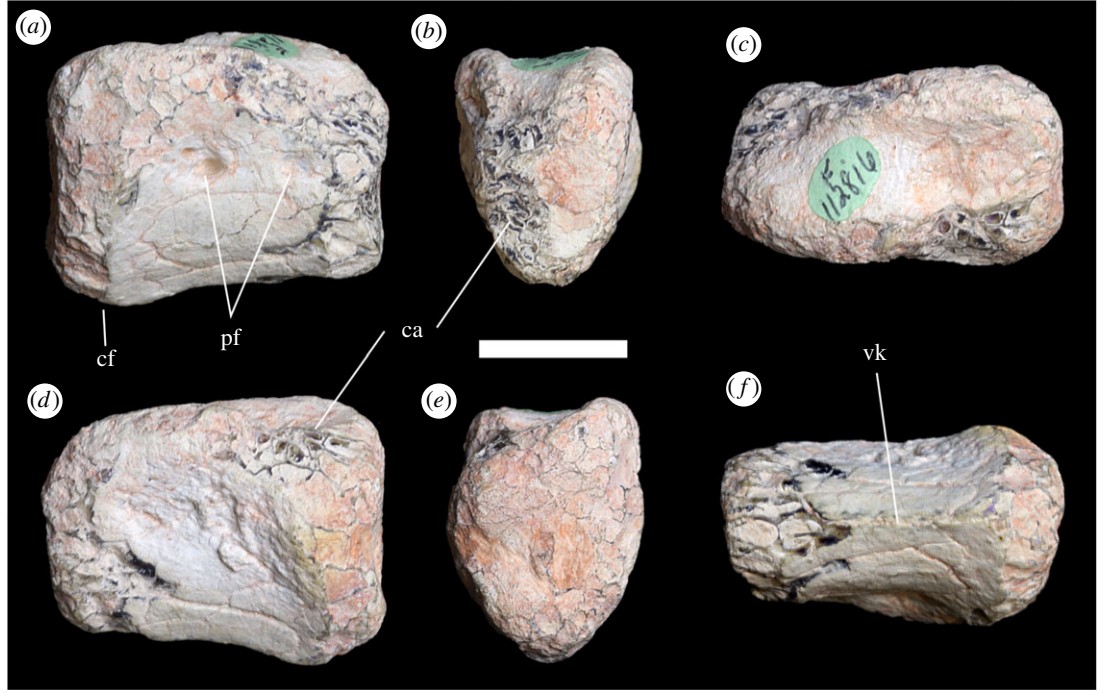

**Figure 8.** Caudal vertebral centrum AM F112816 in (*a*) right lateral, (*b*) anterior, (*c*) dorsal, (*d*) left lateral, (*e*) posterior and (*f*) ventral views. ca, camellae; cf, chevron facet; pf, pneumatic foramina; vk, ventral keel. Scale bar equals 20 mm.

with preserved axial and pelvic elements of comparable size to LRF 3310–3312 are the basal ornithopod *Muttaburrasaurus langdoni* from the Albian Mackunda Formation [58] and the titanosauriform sauropods *Wintonotitan wattsi* [9,59] and *Savannasaurus elliottorum* [60] from the upper Albian–Turonian Winton Formation, all from Queensland. The anterior caudal vertebrae of *Muttaburrasaurus* differ from LRF 3310 and LRF 3311 (aside from the lack of camellate internal composition of the centrum in the former) in the considerably less well-developed mediolateral compression of the centra at mid-length, and the presence of paired ventrolateral ridges on the centrum that extend between the anterior articular facet and the posterior chevron facet, delimiting a well-defined midline groove [58]. The anterior caudal centra of *Wintonotitan* have anterior and posterior articular surfaces that are dorsoventrally compressed, a feature common to most neosauropods [59] and unlike the subcircular surfaces of LRF 3310 and LRF 3312. In addition, *Wintonotitan* has anterior caudal vertebrae with a solid internal texture and that lack a hyposphene–hypantrum system [59]; this contrasts, respectively, with the presence of camellate internal structure and a hyposphene in LRF 3310. The anterior-most caudal centra of *Savannasaurus* have pneumatic fossae [60], a feature that is absent on LRF 3310 and LRF 3311. The robust trapezoidal element LRF 3312 does not resemble part of any of the peduncles, or any other part of, the pelvic elements of *Muttaburrasaurus* [58], and is unlike the more rounded pubic and ischial articulations of the ilium in *Diamantinasaurus* [61].

A position within the caudal series of the vertebral column for LRF 3310 is inferred based on the weak development of laminae and fossae, contrasting with the cervical and dorsal vertebrae of theropods and sauropomomorphs where these features are better developed [55] and the lack of any evidence of sacral rib attachments. Within the caudal series, the presence of a robust transverse process, the anteroposteriorly short centrum in relation to its dorsoventral height and the absence of a facet for the articulation of the chevron implies an anteriormost position. Similarly, LRF 3311 is identified as an anterior caudal vertebra—in a more posterior position than LRF 3310—based on the presence of a posterior chevron facet; the isolated centrum AM F112816 is identified as a caudal vertebra for the same reason. As mentioned previously, LRF 3310–3312 are considered to belong to a single individual. The mediolateral width of the pubic peduncle of the ilium (LRF 3312) relative to the mediolateral width of the posterior face of the anteriormost caudal centrum (LRF 3310) is within the range of variation when compared with other averostran theropods (table 2), providing additional justification for the close association of the aforementioned elements. The incomplete preservation of AM F106525 precludes an accurate determination of its placement within the vertebral column.

**Table 2.** Measurements of the pubic peduncle of the ilium and anteriormost caudal vertebrae for selected neotheropod taxa. Asterisks indicate values estimated from published figures in the provided sources.

| taxon | source | iliac pubic peduncle | | | caudal vertebra 1 | | ilium pubic peduncle ÷ caudal vertebra 1 width |
| | | length | width | length ÷ width | posterior width | posterior height | |
| --- | --- | --- | --- | --- | --- | --- | --- |
| LRF 3310−3312 | — | 152 | 78 | 1.94 | 101.6 | 93 | 0.77 |
| Aerosteon | [62] | 169 | 81 | 2.09 | 128 | 118 | 0.63 |
| Ichthyovenator | [63] | 138 | 75 | 1.84 | 141 | 120 | 0.53 |
| Majungasaurus | [64,65] | 82* | 62* | 1.32* | 52.3 | 58.8 | 1.19 |
| Neovenator | [66] | 135 | 67 | 2.01 | 100* | 114 | 0.67 |
| Sinraptor dongi | [67] | 130 | 80 | 1.63 | 116 | 94* | 0.69 |

The rounded ventral surface of LRF 3310 contrasts with the keeled ventral surface of LRF 3311. In some theropods, the anteriormost caudals have a flattened or gently curved ventral surface that differs from the condition in more posterior caudals in which a ventral groove or keel may be present [66,67]. A single ventral keel in the anterior caudal vertebrae is present in the allosauroid Neovenator and the megaraptorids Aerosteon and Orkoraptor (F. E. Novas. pers comm.) and was hypothesized to represent a synapomorphy of Neovenatoridae [49]; however, this character also appears in carcharodontosaurids, abelisaurids and megalosaurids [49,68,69]. The anterior caudal vertebrae of most other theropods bear a ventral groove bounded by well-defined ridges [56,64,67,70,71], while others lack either a keel or a groove [72].

The small circular central convexity on the posterior articular surfaces of LRF 3310 and AM F106525 appear to represent a genuine feature and not a taphonomic artefact. In both centra, the convexity occupies a similar proportion of the mediolateral width of the articular surface (approx. 31% and 27%, respectively). Within theropods, similar convex features have been described in megalosauroid tetanurans: the posterior surface of the 12th dorsal vertebra of the Early Cretaceous Asian spinosaurid Ichthyovenator [63]; and the posterior surface of an anterior caudal centrum of the Late Jurassic European megalosaurid Torvosaurus gurneyi [73]. In both taxa, the edges of the convex feature are continuous with the surrounding articular surface as opposed to distinctly elevated from the articular surface as in LRF 3310. Furthermore, the convex features of Ichthyovenator and T. gurneyi are more distinctly domed in comparison to the features on LRF 3310 and AM F106525. The variable development (in terms of their relative size and shape) of convex features on centrum articular surfaces advocates for their independent development in each taxon; therefore, this feature is likely to be of little diagnostic utility. As this feature is uncommon within theropods, its presence in both LRF 3310 and AM F106525 is tentatively interpreted as an indication of close affinities between the two specimens.

Anterior caudal neural spines that are mediolaterally narrow and anteroposteriorly short, restricted to the posterior part of the neural arch and extend to or partially overhang the posterior articular surface, are present in many tetanuran theropods, including megalosauroids [70,74], allosauroids [66,75] and coelurosaurs [72,76]. This differs from the spinosaurid condition in which the anteriormost caudals support the distal part of the sail and have robust neural spines that approach the anteroposterior length of their respective centra [77]. As a consequence of the posterior position of the neural spine, many of the aforementioned taxa also have postzygapophyses that are situated at the base of the neural spine and overhang the posterior articular surface [72]. In most theropods, the anteriormost caudals have postzygapophyseal facets that are angled at greater than $40°$ from the horizontal [56,78,79]; however, a few taxa—including LRF 3310−3312—have postzygapophyseal facets that are angled more shallowly, or lie essentially horizontally [72,75,80,81].

Accessory hyposphene−hypantrum articulations are present in the anterior caudal series of many theropods, particularly basal tetanurans [66,67,72,79,80,82,83] and LRF 3310−3312, but are absent in many theropod lineages [56,64,75]. In abelisaurids and tyrannosauroids, hyposphene-hypantrum

accessory articulations are well developed in the anterior caudals, and may extend into the mid-caudal series [69,72,84].

In LRF 3310, the preserved base of the transverse process projects essentially horizontally from the neural arch, a characteristic of most theropods with the exception of abelisaurids and some basal tetanurans in which the transverse processes of caudal vertebrae typically are inclined from the horizontal by at least 20°, and sometimes as much as 40° or more [69]. The transverse processes of the caudal vertebrae in therizinosaurids are distinctly ventrolaterally oriented [85].

An internal structure of the vertebrae consisting of a large number of irregularly shaped chambers delimited by thin septa is termed camellate [86] and is present in some ceratosaurs [87], carcharodontosaurids [75], megaraptorids [51,62], Neovenator [66] and coelurosaurs [88]. However, some [megaraptorid] theropods may also present vertebrae with the plesiomorphic condition of a smaller number of larger chambers with thicker septa, defined as a camerate structure and present predominantly in basal tetanurans [86,89]. In these taxa, camerate and camellate internal structure may be present within the centrum and neural arch of a single vertebra, respectively [90], or serially within the vertebral centra of a single individual [51]. The internal structure of LRF 3310 and AM F112816 differs from that of AM F106525 in that the centrum of the former two are at least partially camellate and composed of small polygonal chambers with thin septa, whereas the preserved centrum of the latter is camerate with large, dorsoventrally elongated chambers with relatively thicker septa. However, variability in the development of internal pneumatic composition is known to vary both between the centrum and neural arch of a single vertebra as well as serially within the vertebral column of theropods [51,91], thus reducing the utility of such characters as accurate indicators of phylogeny.

Within the axial skeleton, unambiguous pneumaticity in the caudal series, in which pneumatic foramina communicate with internal chambers of the centra, is observed in megaraptorans [51,62], the megalosaurid Torvosaurus tanneri [92], the carcharodontosaurid Carcharodontosaurus [81] and some coelurosaurs [2]. Among neotheropods, axial pneumaticity is ancestrally present in the postaxial cervical and anterior dorsal vertebrae and represents the 'common pattern' [2]. This may be augmented to form the 'extended pattern' by the uninterrupted progression of unambiguous pneumatic features anteriorly into the atlas-axis and/or posteriorly into the posterior dorsal, sacral and caudal vertebrae [2]. In AM F112816, both pneumatic foramina and camellae are present, this indicating the presence of unambiguous pneumaticity in a mid-caudal centrum. Pnematicity extending posteriorly to the middle caudal centra has so far only been reported in oviraptorosaurs [93–95] and megaraptorids [62,96]; however, some proximal caudal centra of Aoniraptor lack pneumatic foramina on one or both sides [96]. In LRF 3310, there is possible evidence for pneumaticity associated with the neural spine, whereas pneumatic fossae or foramina are absent from the lateral surfaces of the centrum. This appears to correspond to the development of axial pneumaticity first in the neural arch followed by the centra as documented in the posteriorward progression of the 'extended pattern' [2]. In addition, from the presence of caudal neural arch pneumaticity it can therefore be inferred that the posterior dorsal and sacral vertebrae of the individual to which LRF 3310 pertains were also pneumatic. Unambiguous pneumatic fossae have been documented associated with the neural spine in the abelisauroid Majungasaurus [64,97]. A fossa at the base of the neural arch of an anterior caudal vertebra from a referred specimen of Acrocanthosaurus was considered to be pneumatic [86]; however, a subsequent study did not identify pneumatic anterior caudal vertebrae in this taxon [2]. Imperforate neural arch fossae in anterior caudals were also described from another specimen of Acrocanthosaurus [75]. Variably developed blind neural spine fossae have also been reported in Alioramus altai, Monolophosaurus and Garudimimus [72,98–100]. Differences in the development of this feature on the anterior caudals of Alioramus altai may imply ontogenetic or individual variation, consistent with apneumatic functions such as the sites of axial musculature attachments or fat deposits [72,97].

The pubic peduncle of the ilium is similar in appearance to that of megaraptorids, in particular the well-preserved ilium of Aerosteon [62]. The lateral surface bears fine parallel striations which are interpreted as the attachment site of connective tissue between the pubic peduncle and the pubis; such striations with this inferred function have also been observed on the pubic peduncles of megaraptorans [18,62]. The peduncle appears to be solid with no indication of pneumaticity visible on the lateral surface or in cross-section on the broken dorsal surface, in contrast with the condition of megaraptorids in which pneumatic chambers penetrate the ilia through the medial surface, the brevis fossa and/or the pubic peduncle [18,62]. The triangular ventral surface of the pubic peduncle has proportions similar to those of Allosaurus, megaraptorans (e.g. Aerosteon, [68]) and basal coelurosaurs

(e.g. *Aniksosaurus*, [101]; *Juratyrant*, Brusatte & Benson [102]). In LRF 3312 and the aforementioned taxa, the anteroposterior length of the pubic peduncle approaches or exceeds twice the mediolateral width. This is similar to that reported for *Allosaurus* and megaraptorids and markedly more elongate than those of ceratosaurians and most megalosauroids, although the proportions of the pubic peduncle of the spinosaurid *Ichthyovenator* approach those of LRF 3312 (table 2).

On the basis of the characters discussed above, LRF 3310–3312 presents a combination of characters that indicate a probable phylogenetic position within Avetheropoda: camellae in the caudal centra, a ventral keel on the anterior caudal centra and a pubic peduncle approximately twice as long anteroposteriorly as mediolaterally wide. Unfortunately, the absence of any synapomorphies in LRF 3310–3312 precludes referral to a particular avetheropodan clade. LRF 3310 and AM F106525 both have in common a central convexity on the articular surface of the centrum, an uncommon feature among theropods. The shared presence of this unusual characteristic in both vertebrae, together with their relative proximity to each other suggests that they may pertain to the same taxon, or similar taxa, of avetheropodan theropod. The differences in the internal composition of LRF 3310 and AM F106525 would not preclude the possibility of a close relationship as variation in vertebral pneumatic features within and between vertebrae has been documented for a megaraptorid [51].

Among Australian megaraptorids, the pubic peduncle of LRF 100–106 bears a pneumatic internal composition; the fragment of the main body of the ilium in *Australovenator* shows evidence of pneumaticity [9], but it is not known if the pubic peduncle was pneumatic. As there is no indication of pneumaticity in LRF 3312, it is therefore distinguishable from LRF 100–106. The absence of any vertebral material from either LRF 100–106 or *Australovenator* unfortunately limits the extent to which any additional direct comparisons between the three taxa can be made. However, LRF 3310–3312 appears to lack the development of both pelvic and caudal pneumaticity present in almost all megaraptorids in which these elements are preserved.

AM F105662 can be interpreted as pertaining to a theropod based on the presence of the deep groove for the ligamentum capitis femoris, which first appeared at the common ancestor of Neotheropoda [57]. Within Neotheropoda, transphyseal striations on the anterior surface of the femoral head, such as those on AM F105662, are present in ceratosaurians (*Ceratosaurus*), allosauroids (*Allosaurus fragilis*, *Tyrannotitan chubutensis*), tyrannosaurids (*Alioramus altai*, *Tyrannosaurus rex*) and ornithomimids [57]. A cartilage cone trough (often referred as an 'articular groove') on the proximal surface of the femoral head is present in basal neotheropods and megalosauroids, such as *Ceratosaurus* [57], *Masiakasaurus* [103], *Cryolophosaurus* [104] and *Eustreptospondylus* [70], but is absent in avetheropods [12,56,67,75,105], and also in AM F105662. In most coelurosaurs, the proximal profile of the femoral head in anterior-posterior view is concave due to the presence of an anteroposteriorly oriented groove separating the capital end of the femoral head from the greater trochanter [106]. In AM F105662, the proximal profile of the femoral head is essentially flat, with no such groove present. AM F105662 is morphologically similar to the right femoral head of *Australovenator* [9,12]; however, *Australovenator* differs in lacking transphyseal striations on the anterior surface and having a stronger reduction in anteroposterior width from the capital to trochanteric end. In summary, the combination of characters presented by AM F105662 is indicative of avetheropodan affinities, and is morphologically very similar to the femoral heads of *Allosaurus fragilis* and *Australovenator*.

AM F112816 most probably pertains to a megaraptoran, primarily on the basis of pneumaticity within a mid-caudal centra. The only other theropod clade in which pneumatic mid-caudal vertebrae have been reported, Oviraptorosauria, is unlikely to be a candidate for the affinities of this centrum as the group is likely to have been entirely absent from Gondwana. Restudy of previous material referred to Oviraptorosauria from South America [107,108] has concluded that they are representative of noasaurids and megaraptorans, respectively [109,110].

To date, a majority of the material of Australian apex theropod predators has been referred to Megaraptora [2,5,18,49]. Recently, the Victorian Otway and Gippsland groups have yielded individual specimens interpreted as representatives of other medium and large-sized theropod clades, including a tyrannosauroid pubis, a spinosaurid cervical vertebra and a ceratosaurian astragalocalcaneum [6–8]. The diagnoses for each of these remains were subsequently disputed, with the tyrannosauroid pubis interpreted as either an indeterminate tetanuran or possible megaraptoran, [13,111], and the spinosaurid and ceratosaurian specimens as indeterminate averostrans [13]. A re-evaluation of the problematic Victorian elements is beyond the scope of this present study; however, the new Lightning Ridge theropod material described here further highlights the preponderance of basal avetheropodan-grade theropods from the Mid-Cretaceous of eastern Australia.

# 7. Conclusion

New associated and isolated material from the Upper Cretaceous Griman Creek Formation of Lightning Ridge contributes to the scant fossil record of Australian theropods and underscores the predominance of basal avetheropodan-grade theropods during this interval. LRF 3310–3312, tentatively grouped together with AM 106525, represents axial and pelvic material of an indeterminate medium-sized theropod of indeterminate avetheropodan affinities, and is only the third Australian theropod recognized from associated material. Similarly, AM F105662 is interpreted as the femoral head of an avetheropodan theropod, and shows particular similarities with the femora of *Allosaurus* and *Australovenator*. AM F112816 is identified as a megaraptorid mid-caudal vertebra due to the presence of pneumatic foramina and camellate internal structure, and is the first reported axial skeletal element of such a theropod recognized from Lightning Ridge.

Data accessibility. The raw data for this project are contained within tables 1 and 2.

Authors' contributions. T.B. conceived of the study, acquired and interpreted raw measurement data, wrote the manuscript and approved the final version of the manuscript for publication; E.T.S. acquired raw measurement data and approved the final version of the manuscript for publication; P.R.B. conceived of the study, assisted in drafting and critically revising the manuscript and approved the final version of the manuscript for publication.

Competing interests. We have no competing interests.

Funding. T.B. was funded by a Research Training Program scholarship. P.R.B. was funded under an Australian Research Council Discovery Early Career Researcher Award (project ID: DE170101325).

Acknowledgements. LRF 3300–3312 were gifted to the Australian Opal Centre by one of the authors (E.S.). Jenni Brammall, Manager of the Australian Opal Centre, is also thanked for allowing access to the specimens and providing resources to facilitate their study while in Lightning Ridge. Matthew McCurry of the Australian Museum is thanked for allowing access to material under his care. We extend our thanks to the handling editor Robert Sansom, and to Fernando Novas, Federico Agnolín and an anonymous reviewer for providing useful comments that improved the quality of the manuscript. We acknowledge the Yuwaalaraay, Yuwaalayaay and Gamilaraay custodians of country in the Lightning Ridge district, and pay our respects to Elders past and present.

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
