## [Reviewer comments · Royal Society Open Science]

Review History

RSOS-171832.R0 (Original submission)

Review form: Reviewer 1 (Fernando Novas)

Is the manuscript scientifically sound in its present form?

No

Are the interpretations and conclusions justified by the results?

No

Is the language acceptable?

Yes

Is it clear how to access all supporting data?

Yes

Do you have any ethical concerns with this paper?

No

Have you any concerns about statistical analyses in this paper?

No

Recommendation?

Reject

Comments to the Author(s)

Dear Sir,

My comments on the ms of reference are as follows (these are the same comments inserted as notes in the corrected pdf):

1. GENERAL COMMENTS: With just few exceptions (e.g., Hocknull et al., 2009) Australian dinosaurs are extremely fragmentary. Present ms offers detailed description of deceptively fragmentary elements, which do not escape from this pervasive problem of the Australian Cretaceous record. In my view, the main epistemological problem related with Australian Cretaceous dinosaurs is that researchers (present and previous ones) use such fragmentary evidence as sole and enough support to coin new evolutionary and paleobiogeographic interpretations. Several papers have been published in recent years (e.g., Benson et al., 2012; Barrett, 2010) which are based on very poor facts. Surprisingly, these hypotheses counter information coming from Argentine Patagonia, a neighbour region of Australia which -up to now- offers the most comprehensive fossil record of Cretaceous dinosaurs for Gondwana. Moreover, such hypotheses counters the best preserved and most informative evidence coming from Australia itself! (Winton Fm.). Present ms unfortunately encase in the approach of using fragmentary material to arrive to provocative conclusions: carcharodontosaurids dominated Cretaceous faunas from Australia. Any of the bones here described offer anatomical information to warrant referral to this theropod subclade. Present authors seem reluctant to follow well-supported paleobiogeographic (i.e., Agnolín et al., 2010) and phylogenetic (e.g., Novas et al., 2013) conclusions into which new discoveries from Australia can be comfortably understood. Present ms gives credit to phylogenetic papers (e.g. Benson et al., 2012) that have been already criticised and its ideas dismissed on the basis of more reliable anatomical and phylogenetic analysis (e.g., Novas et al., 2013; Porfiri et al., 2014).
2. Authors express along the ms that "megaraptorids are allosauroids", However, several recent papers support megaraptoran theropods as part of a basal coelurosaurian radiation (e.g., Novas et al., 2013; Porfiri et al., 2014; Novas et al. 2016).
3. The main problem with the present paper is the fossil evidence on which systematic referral and evolutionary interpretations are made. Such fossil evidence consists in numerically scarce, anatomically isolated, and fragmentary preserved bones that, in my view, can not be referred beyond the level of Theropoda indet. No unique derived features can be identified in so fragmentary evidence, and even recognition of features is not warranty to refer such isolated pieces to any particular theropod subgroup.
4. Current available evidence support Megaraptoridae as basal coelurosaurs, not as members of Neovenatoridae. Present authors follow hypothesis published in 2010, instead of more robust analyses on Megaraptoridae relationships published in more recent years.
5. Please, cite: 1) Porfiri et al 2014, Porfiri, J.D., Novas, F.E., Calvo, J.O., Agnolin, F.L., Ezcurra, M.D. and Cerda, I.A. 2014. Juvenile specimen of Megaraptor (Dinosauria, Theropoda) sheds light about tyrannosauroid radiation. *Cretaceous Research* 51; 35-55. And 2) Novas, Agnolin and Aranciaga. 2016. "Phylogenetic relationships of the Cretaceous Gondwanan theropods Megaraptor and Australovenator: the evidence afforded by their manual anatomy". *Memoirs of Museum Victoria* 74: 49-61.
6. Recent analyses depict Megaraptidae as well nested within Tyrannosauroidea.
7. On page 2 authors say that "...fossil discoveries over the past decade, which, when coupled with more comprehensive theropod phylogenetic hypotheses, has yielded new insights into the

diversity of Cretaceous theropods in Australia". However, understanding of poorly documented Australian Cretaceous dinosaur faunas have been obscured by the recurrent (and wrong) view that they are related with Laurasian clades. In fact, Patagonian dinosaurs are key to understand the Australian record. Insights came not from "more comprehensive theropod phylogenetic hypotheses", but on the process of comparing eastern Gondwanan dinosaurs with those of western Gondwana.

8. On page 2 it is said: "The recognition of Megaraptora as a distinct clade of carcharodontosaurian theropods [15] allowed for the first time an unambiguous diagnosis of many problematic Australian theropod specimens". This statement is incorrect: first of all, megaraptorans are not carcharodontosaurians; second, mtc I of Rapator have been reinterpreted as a probable megaraptoran by Agnolín et al. (2010).

9. On page 3 authors state that the new fossils "highlights particular similarities with the carcharodontosaurian rich fauna of Patagonia.". Let me clarify two aspects of this phrase: Lower and Upper Cretaceous theropod faunas from Patagonia include a high diversity of theropods, and probably abelisaurids are the most common. Carcharodontosaurids are recorded in Aptian through Cenomanian rocks. Besides, Cretaceous Cenomanian dinosaur faunas from Africa also include carcharodontosaurid remains, were they seem abundant, alongside with spinosaurids.

10. One of the problems with the present ms is that authors automatically assume that available bones belong to Theropoda. Australia yielded several ornithomimid remains, thus bones here presented require comparisons with these ornithomimids.

11. Authors defend that available bones belong to a single individual. What about the size of these three elements? They keep the same relative proportions seen in more complete allosauroid skeletons?

12. Authors pay special attention to silica deposition on bone surface. However, the almost straight veins of silica, most of which encounter at 90 degrees, suggest a diagenetic origin which do not correspond with the internal histological structure of the bone. In sharp contrast, in well-preserved theropod vertebrae the camera or camella are delimited by thin BONE laminae which do not encounter at 90 degrees. In sum, these are diagenetic crests of silica devoid of both anatomical and phylogenetic value.

13. Regarding on the camerate or camellate condition of theropod vertebrae, Porfiri et al (2014) said: "We demonstrate here that the pneumatic condition of the outer portion of the vertebrae is not uniformly present in the rest of the bone. Thus, studies on the internal structure of theropod vertebrae need to take into account this variation in the degree of pneumaticity, avoiding observations based on localized portions of the bone." Thus, the observed camellate condition in a minimal portion of the vertebra does not warrant the systematic reference of this bone either to Carcharodontosauridae or any other theropod group.

14. On page 10, authors say that "The subtriangular pubic peduncle of the ilium is similar in appearance to those of other partially preserved ilia from megaraptoran allosauroids." However, far from being "partially preserved", the ilia of both Aerosteon and Murusraptor (Coria and Currie 2016) are almost complete and anatomically highly informative. I recommend to compare the isolated bone fragment here described with the pubic pedicles of these two megaraptorans. Besides, let me discourage comparisons with the poorly preserved bone fragment interpreted by Bell et al. as a megaraptorid pubic pedicle.

15. Regarding the internal structure of the pubic pedicle let me ask if this element is massive and entirely devoid of internal structure? If this is true, then think about diagenetic processes affecting the original bone structure.

16. Authors use length/width ratio for the pubic pedicle as enough anatomical information to evaluate to which theropod group this isolated portion of ilium pertains. Nevertheless, I suggest caution with such simple proportions. Osborn (1917) described beautifully preserved Tyrannosaurus ilia in which variations in pubic pedicle proportions (in ventral view) become evident.

17. Authors say "LRF 3310-3312 can be confidently assigned to Carcharodontosauria". But there is

no confidence in referring these badly preserved elements beyond Theropoda.

18. It is also expressed that "Ingroup relations of LRF 3310–3312 within Carcharodontosauria are uncertain". Even accepting the interpretation that bones here described do not belong to Megaraptoridae, why they must be referred as to Carcharodontosauria? Why not another allosauroid or neovenatorid group?

19. On page 11, authors says that "The incomplete fossil record of basal carcharodontosaurids, incomplete and/or poor preservation of known taxa..., and a lack of comprehensive osteological studies of important and relatively complete carcharodontosaurian taxa (e.g. Giganotosaurus, Concavenator) precludes an accurate determination of the extent and development of axial and appendicular pneumaticity within Carcharodontosauria". Please, do not charge against these fossils and their respective descriptions as main causes forbidding the elucidation of Australian fossils. Bones here described are highly fragmentary and devoid of trustable anatomical information, and this is the reason they can't be confidently assigned to any theropod clade.

20. On page 13, it is established that "Therefore, LRF 972 most likely pertains to a theropod.". However, I invite to review this assignment after comparing this vertebra with anterior dorsals of ornithomimid dinosaurs. Please, take a look at Galton 1981 description on Dryosaurus.

21. In some parts of the ms, authors dedicate to describe the "state of art" of different aspects of theropod anatomy. On page 14, they analyse in some depth the systematic position of different avialan taxa. But these thoughts are irrelevant here, because they do not contribute to elucidate the taxonomic referral of the Australian bones.

22. In discussing the "Comments on Australian theropod diversity" I must remember that we have published a comprehensive review on the Cretaceous theropods from Australia (Novas et al. 2013). Although this reference is cited in the present ms, the hypotheses expressed in our paper are neither mentioned, supported or dismissed in the present ms.

23. Authors cite that "Spinosaurid and ceratosaurian theropods have been reported on the basis of isolated elements from the Early Cretaceous of Victoria [2,4,5]." But the purported presence in Australia of these two clades has been dismissed by Novas et al. 2013.

24. On page 18, authors express that " During the Cretaceous, abelisauroids inhabited the lower palaeolatitudes of Gondwana, which were characterised by warm and arid environments [21], whereas cooler conditions were experienced in Australia due to its relatively higher palaeolatitude. During the Cretaceous, abelisauroids inhabited the lower palaeolatitudes of Gondwana, which were characterised by warm and arid environments [21], whereas cooler conditions were experienced in Australia due to its relatively higher palaeolatitude". On this regard I must say that Gondwanan abelisauroids are recorded from low (i.e., Morocco, Egypt) through high (Santa Cruz Province, Patagonia, Argentina) paleolatitudes. This later location occupied a similar paleolatitude as Winton, Australia, for example.

25. I am prone to assume that Australia had different paleoenvironmental conditions with respect to other Gondwanan regions. However, my concern here is with paleolatitudes. I suggest present author to check if Australian fossil sites were or not at similar paleolatitudes of productive dinosaur localities in southern Patagonia.

26. As a conclusion, authors say "In summary, Australia's carcharodontosaurian-dominated theropod fauna bears the closest similarity to that of South America." Contrary to this assertion, available fossil record still suggests Australia as megaraptorid-dominated theropod fauna, by the way similar to Patagonia.

In sum, I consider the evidence presented in the ms as inconclusive and highly fragmentary as to propose novel interpretations on Australian dinosaur diversity.

Sincerely yours,

Fernando E. Novas

Museo Argentino de Ciencias Naturales, Buenos Aires

Review form: Reviewer 2

Is the manuscript scientifically sound in its present form?

No

Are the interpretations and conclusions justified by the results?

No

Is the language acceptable?

Yes

Is it clear how to access all supporting data?

Not Applicable

Do you have any ethical concerns with this paper?

No

Have you any concerns about statistical analyses in this paper?

No

Recommendation?

Major revision is needed (please make suggestions in comments)

Comments to the Author(s)

The fossils are new, and the authors make some good points that suggest they may not belong to megaraptorans (unlike most previous theropod remains from Australia). The descriptive elements of the paper are largely publishable and are certainly well-written. The authors have done a good job with the breadth and precision of their comparisons.

On the other hand, I don't agree with the broader interpretations of significance as currently presented. To summarise, the authors seem to say that finding a non-megaraptoran carcharodontosaurian in Australia conflicts with some previous biogeography hypotheses, and suggests a 'Gondwana-like' (meaning, central/northern South America-like) faunal composition. There are two reasons why I disagree with this:

(1) Gondwana isn't homogeneous, and there is clear evidence for changes in the abundances of higher taxa with latitude. In particular, southern Australian and Antarctic assemblages are rich in small-bodied ornithischians, and poor in sauropods and abelisauroids. This is the claim of various previous works, and I don't think the authors' findings really conflict with that. But because they don't really discuss the hypotheses with sufficient nuance this is basically glossed over.

(2) Basically all large-bodied theropods found globally in the Early Cretaceous - early Late Cretaceous in both Laurasia and Gondwana are carcharodontosaurians (with a small smattering of spinosaurids). In fact, I suspect there are more species of carcharodontosaurians known from Laurasia than Gondwana. So it isn't right to say that finding carcharodontosaurians in Australia indicates 'Gondwanan signal'. At least, not in the absence of good information on their affinities within Carcharodontosauria,

(3) The idea of 'Gondwana signal' is vague, and doesn't take into account biogeographic processes like vicariance, dispersal, regional extinction, in situ diversification etc. This is important. Higher taxa (e.g. Carcharodontosauria) can have wide distributions due to ancient

origins, and this is somewhat independent of the more recent biogeographic events that cladistic biogeography attempts to estimate. Much of the authors perceived 'disagreements' with some previous work basically results from the absence of consideration of this.

So, basically I find that the authors are trying to make a 'big story' out of not enough evidence, and with great misrepresentation of previous hypotheses. I'd advocate just removing all this stuff and reporting the bones.

My detailed comments are below.

Abstract

"The newly expanded carcharodontosaurian fauna in Australia existed penecontemporaneously with the peak diversity of the clade in South America and demonstrates an increasingly Gondwana signal in Australia's theropod fauna".

The idea of 'Gondwanan signal' is vague in biogeographical terms. And the recognition of this based on the the occurrence of a non-megaraptoran carcharodontosaurians doesn't follow: non-megaraptoran carcharodontosaurians such as *Acrocanthosaurus*, *Neovenator*, and *Shaochilong* are known from Laurasia, plus *Concavenator*, *Siats*, the list goes on. This is important, because non-megaraptoran carcharodontosaurians essentially constitute nearly all large-bodied Early Cretaceous theropods from Laurasia. Early Cretaceous of Laurasia hasn't yielded many large-bodied theropod fossils. But this is basically good evidence that they were abundant there, and there is no basis to suggest that finding one in the Early Cretaceous of Australia contributes to biogeographic debates.

Introduction

"With respect to non-avian theropods, Cretaceous Australia appears to be dominated by megaraptorid allosauroids [3,8-11], with purported ceratosaurs, spinosaurids and coelurosaurs comprising a smaller proportion of the theropod diversity [2-4]"

It's not sufficient to say 'purported' and leave it hanging. For example, everyone agrees there are coelurosaurs or some sort or another surely? To me, also, the ceratosaur astragalus is decisive, Fitzgerald defended this in some detail and it hasn't been contested since. It's actually one of the more convincing identifications among the whole assemblage.

"The abundance of megaraptorids in both South America and Australia during the Late Cretaceous has been hypothesised to support a Gondwanan influence on the composition of Australia's theropod fauna during at least the Early Cretaceous. This sea is supported by recent palaeographic modelling...However, an alternative hypothesis suggests that a high diversity of theropods in southern Australia, including traditionally Laurasian forms such as dromaeosaurids and tyrannosauroids, resulted from the establishment of a global cosmopolitanism of theropods in the Early Cretaceous, followed by an episode of climate-driven cosmopolitanism".

>I don't really see these as 'alternative hypotheses' even some of the earliest work on Gondwanan biogeography (by Bonaparte and Bonaparte & Kielan-Jarowska) attributes some clades to ancient divergence during Pangean times and some to more recent, in situ events. Also, the occurrence of latitudinal zonation (your 'climate-driven cosmopolitanism') is not inconsistent with the idea that Australia has many 'Gondwana' clades. I believe that the idea is that higher taxa have essentially 'global' distributions, and their abundances within Gondwana/Laurasia could be related to climate. This would explain why southern Australia and

Antarctica have abundant small-bodied ornithischians, for example, and maybe why southern Australia has relatively abundant coelurosaurs remains compared to e.g. Patagonia and Brazil. Advocates of the 'Gondwana fauna' hypotheses more recently have tended to gloss over this in favour of a more simplistic view that all this stuff belongs to special Gondwana clades. To me, it doesn't make sense to assert this strong, end-member possibility at the expense of any nuance of complexity.

Discussion

"The apparent bias of the Australian Cretaceous theropod record towards carcharodontosaurian theropods... has been presented as evidence for provincialism of Australia's theropod fauna [3,5]"

Neither of the cited references makes this assertion in the way it is framed here. Furthermore, statements later in the discussion imply that the above statement is incorrect. But in fact, they are all totally consistent with each other. It sounds like we basically all agree that megaraptorans particularly are strikingly abundant in Australia, consistent with some provincialism in terms of relative abundances, but inconsistent with the statement that carcharodontosaurians attained their "peak abundance in South America during the mid-late Cretaceous" (in fact, it is in the Early Cretaceous of Australia...), and consistent with the statement that "Australia played an active role in the evolution and radiation of Gondwanan megaraptorids."

"Abelisauroids, which formed a significant component of the theropod fauna of the mid-Late Cretaceous of South America, are conspicuously absent in Australia"

This is not correct Fitzgerald et al. (2012) suggested the debated astragalus to be a ceratosaur (with strong evidence) and possibly an abelisauroid. So it's hard to defend the statement from the manuscript that suggests there is positive evidence for the absence of abelisauroids. I certainly wouldn't say they were 'conspicuously absent'. And in fact, they don't become particularly abundant in South America until the Late Cretaceous. So they could easily be undetected at low levels of sampling as in Australia. All we really know is that they occurred at most, at low abundance in the southern Australian assemblage.

"It has also been proposed that Australia's Early Cretaceous theropod community originated in the southern part of Australia following a period of global theropod cosmopolitanism [2,3]"

This is a mis-reading of what those papers [2,3] proposed. The papers specifically discussed the composition of high-latitude assemblages in Australia, and did not say that this gave rise to the biota of lower latitudes. In fact, they seem to discuss the higher latitude assemblage as a separate entity. This also occurs in southern Patagonia and Antarctica, which are richer in small-bodied ornithischians (i.e. more similar to southern Australia) than other parts of South America [discussed in ref. 3].

"Furthermore, Australia appears to have played an active role in the evolution and radiation of Gondwanan megaraptorids, as opposed to acting as an endpoint in theropod geographic evolution [11]"

No-one has proposed that Australia "acted as an endpoint in theropod geographic evolution". This is a straw man.

"In summary, Australia's carcharodontosaurian-dominated theropod fauna bears the closest similarity to that of South America. Although its taxonomic position within Carcharodontosauria

cannot be constrained with certainty... further emphasises the influence of a 'Gondwanan' theropod fauna on Australia".

For reasons discussed at the start of this review, finding a carcharodontosaurian doesn't lend particular support either to Gondwanan or Laurasian 'affinities'.

Decision letter (RSOS-171832.R0)

05-Jan-2018

Dear Mr Brougham:

Manuscript ID RSOS-171832 entitled "A carcharodontosaurian-dominated Australian theropod fauna from the mid-Cretaceous Grimman Creek Formation (Lightning Ridge, New South Wales)" which you submitted to Royal Society Open Science, has been reviewed. The comments from reviewers are included at the bottom of this letter.

In view of the criticisms of the reviewers, the manuscript has been rejected in its current form. However, a new manuscript may be submitted which takes into consideration these comments.

Please note that resubmitting your manuscript does not guarantee eventual acceptance, and that your resubmission will be subject to peer review before a decision is made.

Your resubmitted manuscript should be submitted by 05-Jul-2018. If you are unable to submit by this date please contact the Editorial Office.

Please note that Royal Society Open Science will introduce article processing charges for all new submissions received from 1 January 2018. Charges will also apply to papers transferred to Royal Society Open Science from other Royal Society Publishing journals, as well as papers submitted as part of our collaboration with the Royal Society of Chemistry (<http://rsos.royalsocietypublishing.org/chemistry>). If your manuscript is submitted and accepted for publication after 1 Jan 2018, you will be asked to pay the article processing charge, unless you request a waiver and this is approved by Royal Society Publishing. You can find out more about the charges at <http://rsos.royalsocietypublishing.org/page/charges>. Should you have any queries, please contact openscience@royalsociety.org.

on behalf of Dr Robert Sansom (Associate Editor) and Jon Blundy (Subject Editor)
 openscience@royalsociety.org

Associate Editor Comments to Author (Dr Robert Sansom):

We thank the authors for submission of this manuscript. We have now received two referee reports and both raise serious concerns regarding the manuscript as it stands. Both are of the opinion that the palaeobiogeographic interpretations and implications of the new finds stretch the available data too far and do not consider other contributing factors or information. The second reviewer is of the opinion that the material is so fragmentary as not to be able to support the interpretations made in the manuscript. If the manuscript could be updated to support those interpretations in light of the explicit synapomorphies that are and are not present in the material, broader reference was made to the alternative phylogenetic solutions raised by reviewer 2, consideration of a wider taxonomic scope of comparison as suggested by reviewer 2 (i.e. not just theropod), and each of the reviewers comments are responded to, then it might be possible to reconsider a resubmitted version of this manuscript for publication, pending further reviews.

Reviewers' Comments to Author:

Reviewer: 1

Comments to the Author(s)

Dear Sir,

My comments on the ms of reference are as follows (these are the same comments inserted as notes in the corrected pdf):

1. GENERAL COMMENTS: With just few exceptions (e.g., Hocknull et al., 2009) Australian dinosaurs are extremely fragmentary. Present ms offers detailed description of deceptively fragmentary elements, which do not escape from this pervasive problem of the Australian Cretaceous record. In my view, the main epistemological problem related with Australian Cretaceous dinosaurs is that researchers (present and previous ones) use such fragmentary evidence as sole and enough support to coin new evolutionary and paleobiogeographic interpretations. Several papers have been published in recent years (e.g., Benson et al., 2012; Barrett, 2010) which are based on very poor facts. Surprisingly, these hypotheses counter information coming from Argentine Patagonia, a neighbour region of Australia which -up to now- offers the most comprehensive fossil record of Cretaceous dinosaurs for Gondwana. Moreover, such hypotheses counters the best preserved and most informative evidence coming from Australia itself! (Winton Fm.). Present ms unfortunately encase in the approach of using fragmentary material to arrive to provocative conclusions: carcharodontosaurids dominated Cretaceous faunas from Australia. Any of the bones here described offer anatomical information to warrant referral to this theropod subclade. Present authors seem reluctant to follow well-supported paleobiogeographic (i.e., Agnolín et al., 2010) and phylogenetic (e.g., Novas et al., 2013) conclusions into which new discoveries from Australia can be comfortably understood. Present ms gives credit to phylogenetic papers (e.g, Benson et al., 2012) that have been already criticised and its ideas dismissed on the basis of more reliable anatomical and phylogenetic analysis (e.g., Novas et al., 2013; Porfiri et al., 2014).
2. Authors express along the ms that "megaraptorids are allosauroids", However, several recent papers support megaraptoran theropods as part of a basal coelurosaurian radiation (e.g., Novas et al., 2013; Porfiri et al., 2014; Novas et al. 2016).
3. The main problem with the present paper is the fossil evidence on which systematic referral and evolutionary interpretations are made. Such fossil evidence consists in numerically scarce, anatomically isolated, and fragmentary preserved bones that, in my view, can not be referred

beyond the level of Theropoda indet. No unique derived features can be identified in so fragmentary evidence, and even recognition of features is not warranty to refer such isolated pieces to any particular theropod subgroup.

4. Current available evidence support Megaraptoridae as basal coelurosaur, not as members of Neovenatoridae. Present authors follow hypothesis published in 2010, instead of more robust analyses on Megaraptoridae relationships published in more recent years.

5. Please, cite: 1) Porfiri et al 2014, Porfiri, J.D., Novas, F.E., Calvo, J.O., Agnolin, F.L., Ezcurra, M.D. and Cerda, I.A. 2014. Juvenile specimen of Megaraptor (Dinosauria, Theropoda) sheds light about tyrannosauroid radiation. *Cretaceous Research* 51; 35–55. And 2) Novas, Agnolin and Aranciaga. 2016. "Phylogenetic relationships of the Cretaceous Gondwanan theropods Megaraptor and Australovenator: the evidence afforded by their manual anatomy". *Memoirs of Museum Victoria* 74: 49–61.

6. Recent analyses depict Megaraptidae as well nested within Tyrannosauroidea.

7. On page 2 authors say that "...fossil discoveries over the past decade, which, when coupled with more comprehensive theropod phylogenetic hypotheses, has yielded new insights into the diversity of Cretaceous theropods in Australia". However, understanding of poorly documented Australian Cretaceous dinosaur faunas have been obscured by the recurrent (and wrong) view that they are related with Laurasian clades. In fact, Patagonian dinosaurs are key to understand the Australian record. Insights came not from "more comprehensive theropod phylogenetic hypotheses", but on the process of comparing eastern Gondwanan dinosaurs with those of western Gondwana.

8. On page 2 it is said: "The recognition of Megaraptora as a distinct clade of carcharodontosaurian theropods [15] allowed for the first time an unambiguous diagnosis of many problematic Australian theropod specimens". This statement is incorrect: first of all, megaraptorans are not carcharodontosaurians; second, mtc I of Rapator have been reinterpreted as a probable megaraptoran by Agnolín et al. (2010).

9. On page 3 authors state that the new fossils "highlights particular similarities with the carcharodontosaurian rich fauna of Patagonia.". Let me clarify two aspects of this phrase: Lower and Upper Cretaceous theropod faunas from Patagonia include a high diversity of theropods, and probably abelisaurids are the most common. Carcharodontosaurids are recorded in Aptian through Cenomanian rocks. Besides, Cretaceous Cenomanian dinosaur faunas from Africa also include carcharodontosaurid remains, were they seem abundant, alongside with spinosaurids.

10. One of the problems with the present ms is that authors automatically assume that available bones belong to Theropoda. Australia yielded several ornithomimid remains, thus bones here presented require comparisons with these ornithomimids.

11. Authors defend that available bones belong to a single individual. What about the size of these three elements? They keep the same relative proportions seen in more complete allosauroid skeletons?

12. Authors pay special attention to silica deposition on bone surface. However, the almost straight veins of silica, most of which encounter at 90 degrees, suggest a diagenetic origin which do not correspond with the internal histological structure of the bone. In sharp contrast, in well-preserved theropod vertebrae the camera or camella are delimited by thin BONE laminae which do not encounter at 90 degrees. In sum, these are diagenetic crests of silica devoid of both anatomical and phylogenetic value.

13. Regarding on the camerate or camellate condition of theropod vertebrae, Porfiri et al (2014) said: "We demonstrate here that the pneumatic condition of the outer portion of the vertebrae is not uniformly present in the rest of the bone. Thus, studies on the internal structure of theropod vertebrae need to take into account this variation in the degree of pneumaticity, avoiding observations based on localized portions of the bone." Thus, the observed camellate condition in a minimal portion of the vertebra does not warrant the systematic reference of this bone either to Carcharodontosauridae or any other theropod group.

14. On page 10, authors say that "The subtriangular pubic peduncle of the ilium is similar in appearance to those of other partially preserved ilia from megaraptoran allosauroids." However, far from being "partially preserved", the ilia of both *Aerosteon* and *Murusraptor* (Coria and Currie 2016) are almost complete and anatomically highly informative. I recommend to compare the isolated bone fragment here described with the pubic pedicles of these two megaraptorans. Besides, let me discourage comparisons with the poorly preserved bone fragment interpreted by Bell et al. as a megaraptorid pubic pedicle.

15. Regarding the internal structure of the pubic pedicle let me ask if this element is massive and entirely devoid of internal structure? If this is true, then think about diagenetic processes affecting the original bone structure.

16. Authors use length/width ratio for the pubic pedicle as enough anatomical information to evaluate to which theropod group this isolated portion of ilium pertains. Nevertheless, I suggest caution with such simple proportions. Osborn (1917) described beautifully preserved *Tyrannosaurus* ilia in which variations in pubic pedicle proportions (in ventral view) become evident.

17. Authors say "LRF 3310–3312 can be confidently assigned to Carcharodontosauria". But there is no confidence in referring these badly preserved elements beyond Theropoda.

18. It is also expressed that "Ingroup relations of LRF 3310–3312 within Carcharodontosauria are uncertain". Even accepting the interpretation that bones here described do not belong to Megaraptoridae, why they must be referred as to Carcharodontosauria? Why not another allosauroid or neovenatorid group?

19. On page 11, authors say that "The incomplete fossil record of basal carcharodontosaurids, incomplete and/or poor preservation of known taxa..., and a lack of comprehensive osteological studies of important and relatively complete carcharodontosaurian taxa (e.g. *Giganotosaurus*, *Concavenator*) precludes an accurate determination of the extent and development of axial and appendicular pneumaticity within Carcharodontosauria". Please, do not charge against these fossils and their respective descriptions as main causes forbidding the elucidation of Australian fossils. Bones here described are highly fragmentary and devoid of trustable anatomical information, and this is the reason they can't be confidently assigned to any theropod clade.

20. On page 13, it is established that "Therefore, LRF 972 most likely pertains to a theropod.". However, I invite to review this assignment after comparing this vertebra with anterior dorsals of ornithomimid dinosaurs. Please, take a look at Galton 1981 description on *Dryosaurus*.

21. In some parts of the ms, authors dedicate to describe the "state of art" of different aspects of theropod anatomy. On page 14, they analyse in some depth the systematic position of different avialan taxa. But these thoughts are irrelevant here, because they do not contribute to elucidate the taxonomic referral of the Australian bones.

22. In discussing the "Comments on Australian theropod diversity" I must remember that we have published a comprehensive review on the Cretaceous theropods from Australia (Novas et al. 2013). Although this reference is cited in the present ms, the hypotheses expressed in our paper are neither mentioned, supported or dismissed in the present ms.

23. Authors cite that "Spinosaurid and ceratosaurian theropods have been reported on the basis of isolated elements from the Early Cretaceous of Victoria [2,4,5]." But the purported presence in Australia of these two clades has been dismissed by Novas et al. 2013.

24. On page 18, authors express that "During the Cretaceous, abelisauroids inhabited the lower palaeolatitudes of Gondwana, which were characterised by warm and arid environments [21], whereas cooler conditions were experienced in Australia due to its relatively higher palaeolatitude. During the Cretaceous, abelisauroids inhabited the lower palaeolatitudes of Gondwana, which were characterised by warm and arid environments [21], whereas cooler conditions were experienced in Australia due to its relatively higher palaeolatitude". On this regard I must say that Gondwanan abelisauroids are recorded from low (i.e., Morocco, Egypt) through high (Santa Cruz Province, Patagonia, Argentina)

paleolatitudes. This later location occupied a similar paleolatitude as Winton, Australia, for example.

25. I am prone to assume that Australia had different paleoenvironmental conditions with respect to other Gondwanan regions. However, my concern here is with paleolatitudes. I suggest present author to check if Australian fossil sites were or not at similar paleolatitudes of productive dinosaur localities in southern Patagonia.

26. As a conclusion, authors say "In summary, Australia's carcharodontosaurian-dominated theropod fauna bears the closest similarity to that of South America." Contrary to this assertion, available fossil record still suggests Australia as megaraptorid-dominated theropod fauna, by the way similar to Patagonia.

In sum, I consider the evidence presented in the ms as inconclusive and highly fragmentary as to propose novel interpretations on Australian dinosaur diversity.

Sincerely yours,

Fernando E. Novas

Museo Argentino de Ciencias Naturales, Buenos Aires

Reviewer: 2

Comments to the Author(s)

The fossils are new, and the authors make some good points that suggest they may not belong to megaraptorans (unlike most previous theropod remains from Australia). The descriptive elements of the paper are largely publishable and are certainly well-written. The authors have done a good job with the breadth and precision of their comparisons.

On the other hand, I don't agree with the broader interpretations of significance as currently presented. To summarise, the authors seem to say that finding a non-megaraptoran carcharodontosaurian in Australia conflicts with some previous biogeography hypotheses, and suggests a 'Gondwana-like' (meaning, central/northern South America-like) faunal composition. There are two reasons why I disagree with this:

(1) Gondwana isn't homogeneous, and there is clear evidence for changes in the abundances of higher taxa with latitude. In particular, southern Australian and Antarctic assemblages are rich in small-bodied ornithischians, and poor in sauropods and abelisauroids. This is the claim of various previous works, and I don't think the authors' findings really conflict with that. But because they don't really discuss the hypotheses with sufficient nuance this is basically glossed over.

(2) Basically all large-bodied theropods found globally in the Early Cretaceous - early Late Cretaceous in both Laurasia and Gondwana are carcharodontosaurians (with a small smattering of spinosaurids). In fact, I suspect there are more species of carcharodontosaurians known from Laurasia than Gondwana. So it isn't right to say that finding carcharodontosaurians in Australia indicates 'Gondwanan signal'. At least, not in the absence of good information on their affinities within Carcharodontosauria,

(3) The idea of 'Gondwana signal' is vague, and doesn't take into account biogeographic processes like vicariance, dispersal, regional extinction, in situ diversification etc. This is important. Higher taxa (e.g. Carcharodontosauria) can have wide distributions due to ancient origins, and this is somewhat independent of the more recent biogeographic events that cladistic biogeography attempts to estimate. Much of the authors perceived 'disagreements' with some previous work basically results from the absence of consideration of this.

So, basically I find that the authors are trying to make a 'big story' out of not enough evidence, and with great misrepresentation of previous hypotheses. I'd advocate just removing all this stuff and reporting the bones.

My detailed comments are below.

Abstract

"The newly expanded carcharodontosaurian fauna in Australia existed penecontemporaneously with the peak diversity of the clade in South America and demonstrates an increasingly Gondwana signal in Australia's theropod fauna".

The idea of 'Gondwanan signal' is vague in biogeographical terms. And the recognition of this based on the the occurrence of a non-megaraptoran carcharodontosaurians doesn't follow: non-megaraptoran carcharodontosaurians such as Acrocanthosaurus, Neovenator, and Shaochilong are known from Laurasia, plus Concavenator, Siats, the list goes on. This is important, because non-megaraptoran carcharodontosaurians essentially constitute nearly all large-bodied Early Cretaceous theropods from Laurasia. Early Cretaceous of Laurasia hasn't yielded many large-bodied theropod fossils. But this is basically good evidence that they were abundant there, and there is no basis to suggest that finding one in the Early Cretaceous of Australia contributes to biogeographic debates.

Introduction

"With respect to non-avian theropods, Cretaceous Australia appears to be dominated by megaraptorid allosauroids [3,8-11], with purported ceratosaurs, spinosaurids and coelurosaurs comprising a smaller proportion of the theropod diversity [2-4]"

It's not sufficient to say 'purported' and leave it hanging. For example, everyone agrees there are coelurosaurs or some sort or another surely? To me, also, the ceratosaur astragalus is decisive, Fitzgerald defended this in some detail and it hasn't been contested since. It's actually one of the more convincing identifications among the whole assemblage.

"The abundance of megaraptorids in both South America and Australia during the Late Cretaceous has been hypothesised to support a Gondwanan influence on the composition of Australia's theropod fauna during at least the Early Cretaceous. This sea is supported by recent palaeographic modelling...However, an alternative hypothesis suggests that a high diversity of theropods in southern Australia, including traditionally Laurasian forms such as dromaeosaurids and tyrannosauroids, resulted from the establishment of a global cosmopolitanism of theropods in the Early Cretaceous, followed by an episode of climate-driven cosmopolitanism".

>I don't really see these as 'alternative hypotheses' even some of the earliest work on Gondwanan biogeography (by Bonaparte and Bonaparte & Kielan-Jarowska) attributes some clades to ancient divergence during Pangean times and some to more recent, in situ events. Also, the occurrence of latitudinal zonation (your 'climate-driven cosmopolitanism') is not inconsistent with the idea that Australia has many 'Gondwana' clades. I believe that the idea is that higher taxa have essentially 'global' distributions, and their abundances within

Gondwana/Laurasia could be related to climate. This would explain why southern Australia and Antarctica have abundant small-bodied ornithischians, for example, and maybe why southern Australia has relatively abundant coelurosaurs remains compared to e.. Patagonia and Brazil. Advocates of the 'Gondwana fauna' hypotheses more recently have tended to gloss over this in favour of a more simplistic view that all this stuff belongs to special Gondwana clades. To me, it doesn't make sense to assert this strong, end-member possibility at the expense of any nuance of complexity.

Discussion

"The apparent bias of the Australian Cretaceous theropod record towards carcharodontosaurian theropods... has been presented as evidence for provincialism of Australia's theropod fauna [3,5]"

Neither of the cited references makes this assertion in the way it is framed here. Furthermore, statements later in the discussion imply that the above statement is incorrect. But in fact, they are all totally consistent with each other. It sounds like we basically all agree that megaraptorans particularly are strikingly abundant in Australia, consistent with some provincialism in terms of relative abundances, but inconsistent with the statement that carcharodontosaurians attained their "peak abundance in South America during the mid-late Cretaceous" (in fact, it is in the Early Cretaceous of Australia...), and consistent with the statement that "Australia played an active role in the evolution and radiation of Gondwanan megaraptorids."

"Abelisauroids, which formed a significant component of the theropod fauna of the mid-Late Cretaceous of South America, are conspicuously absent in Australia"

This is not correct Fitzgerald et al. (2012) suggested the debated astragalus to be a ceratosaur (with strong evidence) and possibly an abelisauroid. So it's hard to defend the statement from the manuscript that suggests there is positive evidence for the absence of abelisauroids. I certainly wouldn't say they were 'conspicuously absent'. And in fact, they don't become particularly abundant in South America until the Late Cretaceous. So they could easily be undetected at low levels of sampling as in Australia. All we really know is that they occurred at most, at low abundance in the southern Australian assemblage.

"It has also been proposed that Australia's Early Cretaceous theropod community originated in the southern part of Australia following a period of global theropod cosmopolitanism [2,3]"

This is a mis-reading of what those papers [2,3] proposed. The papers specifically discussed the composition of high-latitude assemblages in Australia, and did not say that this gave rise to the biota of lower latitudes. In fact, they seem to discuss the higher latitude assemblage as a separate entity. This also occurs in southern Patagonia and Antarctica, which are richer in small-bodied ornithischians (i.e. more similar to southern Australia) than other parts of South America [discussed in ref. 3].

"Furthermore, Australia appears to have played an active role in the evolution and radiation of Gondwanan megaraptorids, as opposed to acting as an endpoint in theropod geographic evolution [11]"

No-one has proposed that Australia “acted as an endpoint in theropod geographic evolution”. This is a straw man.

“In summary, Australia’s carcharodontosaurian-dominated theropod fauna bears the closest similarity to that of South America. Although its taxonomic position within Carcharodontosauria cannot be constrained with certainty... further emphasises the influence of a ‘Gondwanan’ theropod fauna on Australia”.

For reasons discussed at the start of this review, finding a carcharodontosaurian doesn’t lend particular support either to Gondwanan or Laurasian ‘affinities’.

Author's Response to Decision Letter for (RSOS-171832.R0)

See Appendix A.

RSOS-180826.R0

Review form: Reviewer 1 (Fernando Novas)

Is the manuscript scientifically sound in its present form?

No

Are the interpretations and conclusions justified by the results?

No

Is the language acceptable?

Yes

Is it clear how to access all supporting data?

Yes

Do you have any ethical concerns with this paper?

No

Have you any concerns about statistical analyses in this paper?

No

Recommendation?

Major revision is needed (please make suggestions in comments)

Comments to the Author(s)

New carcharodontosaurian theropod remains from the mid-Cretaceous Griman Creek Formation, Lightning Ridge (New South Wales, Australia)

1. Present paper is well presented, materials are well described and compared, and bibliography

is updated. However, the methodological approach has some discussable aspects: first, the fragmentary preservation of the scarce available bones constitute a serious obstacle for firm taxonomic referral; 2) novel conclusions on theropod faunal composition presented by the authors are based on fragmentary and poorly informative bones; 3) main taxonomic, phylogenetic and paleobiogeographic conclusions are framed within interpretations made by Benson et al. 2010, which analysed character data and theropod taxa in a partial way, overlooking apomorphic similarities that megaraptorans share with coelurosaurs in general, and tyrannosauroids in particular. In sum, present paper, in my view, reports on new theropod remains which neither amplify nor modify current interpretations on the taxonomic composition of theropod Cretaceous faunas from Australia.

2. When referring to Patagonian theropod faunas, let me suggest something like: "...at roughly the same palaeolatitude, which hosted a diverse range of abelisaurids, alongside carcharodontosaurids and megaraptorids". The reason of this change in the phrase is that in Patagonia the numerically dominant theropods (in the lapse Aptian through Turonian) are abelisaurids, being seconded by carcharodontosaurids and megaraptorids (in this order). Thus, the main difference between Australia and Patagonia is the predominance of megaraptorids in the first continent, vs abelisaurids in the second one (currently unrecorded in Australia).

3. The use of "carcharodontosaurian" to describe the Australian theropod faunas is misleading, and I emphatically recommend dismiss its use (based on what we currently know about Australia fossil record). The reasons are two: first, megaraptorids are, by far, the most frequently found theropods in Australia; second, the term "carcharodontosaurian" that is used to gather carcharodontosaurids plus megaraptorids, is by following Benson et al 2010, a hypothesis that is weaker than that depicting megaraptorids as coelurosaurs (as detailed in Novas et al., 2013, Porfiri et al. 2014, and more recently by a different crew leaded by Porfiri et al. 2018).

4. The systematic framework, as here exposed, arbitrarily takes part for an already contested hypothesis on megaraptoran relationships. Present authors do not follow alternative hypotheses (i.e., Novas et al., 2013; Porfiri et al., 2014, 2018) based on a wrong argument: they state about the necessity to "incorporate a broader sampling of basal tetanurans and basal coelurosaurs characters and taxa is required before either hypothesis can be accepted over the current consensus view". But we have ALREADY DID this task! In Novas et al. 2013 paper, we incorporated a broader sampling than that presented by Benson, by merging two comprehensive datasets: the one by Benson et al. 2010 on basal tetanurans, and the one by Brusatte et al. 2010 on tyrannosauroids. With the incorporation of these later theropods the results got were substantially different from those exposed by Benson et al 2010. In other words, present authors counters their own proposal to "incorporate a broader sampling of characters and taxa", by choosing the more restrictive dataset (i.e., Benson's dataset). By the way, the same criticism applies for Brusatte and Carr (2016; ref 52 of the present ms), who overlooked our methodological approach. Finally, it is clear that there is no "current consensus" that megaraptorans are neovenatorid carcharodontosaurians: a recent paper by Porfiri et al. 2018 (a different working group that mine) concluded that megaraptorans are coelurosaurs, demonstrating, again, that the old idea that megaraptorans are allosauroids has not the best support.

5. Countering present authors, the evidence presented here is ambiguous -at least- to support referral to Allosauroidea and Carcharodontosauria / Carcharodontosauridae. The available bones can be also referred to Megaraptoridae, based on morphology, being in accordance with previous discoveries of the same group of theropods in the same fossil site as well as other sites in Australia.

6. I strongly discourage determination of the kind and degree of internal structure of vertebra just observing limited (and not analogous) portions of a vertebra. To get reliable information on this

yet poorly known aspect of theropod anatomy, it is needed to make comparisons based on equivalent cross-section planes. Porfiri et al 2014 called attention on this aspect by documenting the presence of camerate and camellate conditions in different parts of a single dorsal centrum of Megaraptor.

7. The Patagonian megaraptorid *Orkoraptor* has a keel on the ventral side of proximal caudal centra. This feature was neither described nor illustrated in its original paper (Novas et al. 2008), but I can provide the authors with images of the vertebra showing this feature. A ventral keel of this kind is also present in a mid-caudal of *Aerosteon*.

8. This length/width ratio applies to many basal tetanurans, including *Allosaurus*, *Giganotosaurus*, *Aerosteon*, *Anyksosaurus*, *Juratyran*. Thus, it does not appear to diagnose a particular tetanuran clade.

9. Based on the comments made above on each of the three listed features, I must conclude that they do not conform a set of characters "commonly found" among carcharodontosaurians, but among basal tetanurans. Again: the problem is not with characters but with the limited factual evidence to discern to which particular taxonomic group they belong.

10. Unfortunately, the evidence yielded from this fossil site is not decisive to certify the presence of a theropod group other than already described megaraptorids.

Fernando E. Novas
Principal Researcher Conicet
Head Laboratory of Comparative Anatomy
Museo Argentino de Ciencias Naturales, Buenos Aires
Argentina

Review form: Reviewer 3 (Federico Agnolin)

Is the manuscript scientifically sound in its present form?

Yes

Are the interpretations and conclusions justified by the results?

Yes

Is the language acceptable?

Yes

Is it clear how to access all supporting data?

Yes

Do you have any ethical concerns with this paper?

No

Have you any concerns about statistical analyses in this paper?

Yes

Recommendation?

Accept with minor revision (please list in comments)

Comments to the Author(s)

Dear Authors and Editor,

I congratulate the authors for such a well-written and concise article. This MS contributes to the knowledge of the still poorly known and enigmatic theropod faunas from Australia. I have some concerns about the identification of the material and the features sustaining it. I think that authors should improve comparisons with carcharodontosaurids and megaraptorans. I include some brief comments in this regard within the PDF that I am attaching (Appendix B).

All the best,

Federico AGnolin

Decision letter (RSOS-180826.R0)

21-Sep-2018

Dear Mr Brougham,

The Subject Editor assigned to your paper ("New carcharodontosaurian theropod remains from the mid-Cretaceous Griman Creek Formation, Lightning Ridge (New South Wales, Australia)") has now received comments from reviewers. We would like you to revise your paper in accordance with the referee and Associate Editor suggestions which can be found below (not including confidential reports to the Editor). Please note this decision does not guarantee eventual acceptance.

Please submit a copy of your revised paper before 14-Oct-2018. Please note that the revision deadline will expire at 00.00am on this date. If we do not hear from you within this time then it will be assumed that the paper has been withdrawn. In exceptional circumstances, extensions may be possible if agreed with the Editorial Office in advance. We do not allow multiple rounds of revision so we urge you to make every effort to fully address all of the comments at this stage. If deemed necessary by the Editors, your manuscript will be sent back to one or more of the original reviewers for assessment. If the original reviewers are not available we may invite new reviewers.

When submitting your revised manuscript, you must respond to the comments made by the referees and upload a file "Response to Referees" in "Section 6 - File Upload". Please use this to document how you have responded to each of the comments, and the adjustments you have made. In order to expedite the processing of the revised manuscript, please be as specific as possible in your response.

- Ethics statement

If your study uses humans or animals please include details of the ethical approval received, including the name of the committee that granted approval. For human studies please also detail

whether informed consent was obtained. For field studies on animals please include details of all permissions, licences and/or approvals granted to carry out the fieldwork.

- Data accessibility

If you wish to submit your supporting data or code to Dryad (<http://datadryad.org/>), or modify your current submission to dryad, please use the following link:
<http://datadryad.org/submit?journalID=RSOS&manu=RSOS-180826>

- Competing interests

- Authors' contributions

- Acknowledgements

- Funding statement

Please note that Royal Society Open Science charge article processing charges for all new submissions that are accepted for publication. Charges will also apply to papers transferred to Royal Society Open Science from other Royal Society Publishing journals, as well as papers submitted as part of our collaboration with the Royal Society of Chemistry (<http://rsos.royalsocietypublishing.org/chemistry>). If your manuscript is newly submitted and subsequently accepted for publication, you will be asked to pay the article processing charge, unless you request a waiver and this is approved by Royal Society Publishing. You can find out

more about the charges at <http://rsos.royalsocietypublishing.org/page/charges>. Should you have any queries, please contact openscience@royalsociety.org.

on behalf of Dr Robert Sansom (Associate Editor) and Prof. Jon Blundy (Subject Editor)
openscience@royalsociety.org

Associate Editor Comments to Author (Dr Robert Sansom):

Associate Editor

Comments to the Author:

The authors have taken care and attention to address the first round of reviewers and the resulting MS is much improved. In response to this revised MS, reviewer 2 raises some minor concerns which should be addressed. Reviewer 1 raises more serious concerns. In the most part they relate to the taxonomic framework. I recommend that the authors carefully consider the review and either explicitly justify the use of the Benson 2010 framework (as oppose to the more recent ones discussed by the reviewer) or follow the alternative taxonomic groups and resulting synapomorphies detailed by reviewer 1. I look forward to seeing the revised manuscript and response letter.

Reviewer comments to Author:

Reviewer: 1

Comments to the Author(s)

New carcharodontosaurian theropod remains from the mid-Cretaceous Griman Creek Formation, Lightning Ridge (New South Wales, Australia)

1. Present paper is well presented, materials are well described and compared, and bibliography is updated. However, the methodological approach has some discussable aspects: first, the fragmentary preservation of the scarce available bones constitute a serious obstacle for firm taxonomic referral; 2) novel conclusions on theropod faunal composition presented by the authors are based on fragmentary and poorly informative bones; 3) main taxonomic, phylogenetic and paleobiogeographic conclusions are framed within interpretations made by Benson et al. 2010, which analysed character data and theropod taxa in a partial way, overlooking apomorphic similarities that megaraptorans share with coelurosaurs in general, and tyrannosauroids in particular. In sum, present paper, in my view, reports on new theropod remains which neither amplify nor modify current interpretations on the taxonomic composition of theropod Cretaceous faunas from Australia.

2. When referring to Patagonian theropod faunas, let me suggest something like: "...at roughly the same palaeolatitude, which hosted a diverse range of abelisaurids, alongside carcharodontosaurids and megaraptorids". The reason of this change in the phrase is that in Patagonia the numerically dominant theropods (in the lapse Aptian through Turonian) are abelisaurids, being seconded by carcharodontosaurids and megaraptorids (in this order). Thus,

the main difference between Australia and Patagonia is the predominance of megaraptorids in the first continent, vs abelisaurids in the second one (currently unrecorded in Australia).

3. The use of "carcharodontosaurian" to describe the Australian theropod faunas is misleading, and I emphatically recommend dismiss its use (based on what we currently know about Australia fossil record). The reasons are two: first, megaraptorids are, by far, the most frequently found theropods in Australia; second, the term "carcharodontosaurian" that is used to gather carcharodontosaurids plus megaraptorids, is by following Benson et al 2010, a hypothesis that is weaker than that depicting megaraptorids as coelurosaurians (as detailed in Novas et al., 2013, Porfiri et al. 2014, and more recently by a different crew led by Porfiri et al. 2018).

4. The systematic framework, as here exposed, arbitrarily takes part for an already contested hypothesis on megaraptoran relationships. Present authors do not follow alternative hypotheses (i.e., Novas et al., 2013; Porfiri et al., 2014, 2018) based on a wrong argument: they state about the necessity to "incorporate a broader sampling of basal tetanurans and basal coelurosaurian characters and taxa is required before either hypothesis can be accepted over the current consensus view". But we have ALREADY DID this task! In Novas et al. 2013 paper, we incorporated a broader sampling than that presented by Benson, by merging two comprehensive datasets: the one by Benson et al. 2010 on basal tetanurans, and the one by Brusatte et al. 2010 on tyrannosauroids. With the incorporation of these later theropods the results got were substantially different from those exposed by Benson et al 2010. In other words, present authors counters their own proposal to "incorporate a broader sampling of characters and taxa", by choosing the more restrictive dataset (i.e., Benson's dataset). By the way, the same criticism applies for Brusatte and Carr (2016; ref 52 of the present ms), who overlooked our methodological approach. Finally, it is clear that there is no "current consensus" that megaraptorans are neovenatorid carcharodontosaurians: a recent paper by Porfiri et al. 2018 (a different working group that mine) concluded that megaraptorans are coelurosaurs, demonstrating, again, that the old idea that megaraptorans are allosauroids has not the best support.

5. Countering present authors, the evidence presented here is ambiguous -at least- to support referral to Allosauroidae and Carcharodontosauria / Carcharodontosauridae. The available bones can be also referred to Megaraptoridae, based on morphology, being in accordance with previous discoveries of the same group of theropods in the same fossil site as well as other sites in Australia.

6. I strongly discourage determination of the kind and degree of internal structure of vertebra just observing limited (and not analogous) portions of a vertebra. To get reliable information on this yet poorly known aspect of theropod anatomy, it is needed to make comparisons based on equivalent cross-section planes. Porfiri et al 2014 called attention on this aspect by documenting the presence of camerate and camellate conditions in different parts of a single dorsal centrum of Megaraptor.

7. The Patagonian megaraptorid Orkoraptor has a keel on the ventral side of proximal caudal centra. This feature was neither described nor illustrated in its original paper (Novas et al. 2008), but I can provide the authors with images of the vertebra showing this feature. A ventral keel of this kind is also present in a mid-caudal of Aerosteon.

8. This length/width ratio applies to many basal tetanurans, including Allosaurus, Giganotosaurus, Aerosteon, Anyksosaurus, Juratyrann. Thus, it does not appear to diagnose a particular tetanuran clade.

9. Based on the comments made above on each of the three listed features, I must conclude that they do not conform a set of characters "commonly found" among carcharodontosaurians, but

among basal tetanurans. Again: the problem is not with characters but with the limited factual evidence to discern to which particular taxonomic group they belong.

10. Unfortunately, the evidence yielded from this fossil site is not decisive to certify the presence of a theropod group other than already described megaraptorids.

Fernando E. Novas
Principal Researcher Conicet
Head Laboratory of Comparative Anatomy
Museo Argentino de Ciencias Naturales, Buenos Aires
Argentina

Reviewer: 3

Comments to the Author(s)

Dear Authors and Editor,

I congratulate the authors for such a well-written and concise article. This MS contributes to the knowledge of the still poorly known and enigmatic theropod faunas from Australia. I have some concerns about the identification of the material and the features sustaining it. I think that authors should improve comparisons with carcharodontosaurids and megaraptorans. I include some brief comments in this regard within the PDDF that I am attaching.

All the best,

Federico Agnolin

Author's Response to Decision Letter for (RSOS-180826.R0)

See Appendix C.

Decision letter (RSOS-180826.R1)

17-Dec-2018

Dear Mr Brougham,

I am pleased to inform you that your manuscript entitled "New theropod material from the mid-Cretaceous Grimman Creek Formation, Lightning Ridge (New South Wales, Australia)" is now accepted for publication in Royal Society Open Science.

Royal Society Open Science operates under a continuous publication model (<http://bit.ly/cpFAQ>). Your article will be published straight into the next open issue and this will be the final version of the paper. As such, it can be cited immediately by other researchers.

As the issue version of your paper will be the only version to be published I would advise you to check your proofs thoroughly as changes cannot be made once the paper is published.

on behalf of Dr Robert Sansom (Associate Editor) and Jon Blundy (Subject Editor)
openscience@royalsociety.org

Associate Editor Comments to Author (Dr Robert Sansom):

We thank the authors for addressing the reviewers comments comprehensively. The final lingering concern of reviewer 1 has been directly addressed in this revised manuscript. Given that, and the positive comments from the other reviewers, I am happy to recommend publication.

Appendix A

Reviewer 1:

Point 1 - The palaeobiogeographic section has been removed. We remain ambivalent with regards to the studies debating the affinities of the Australian theropod specimens raised by Reviewer 1 as this topic is beyond the scope of the present manuscript. In addition, an in-depth consideration of alternative phylogenetic hypotheses is not warranted in the present manuscript, which is purely descriptive in nature. We also express doubts on Reviewer 1's assertions of the purported robustness of the studies upon which the aforementioned alternative hypotheses are ultimately based (see below).

Points 2, 4, 6, 8 - A paragraph stating the reasons for using the preferred phylogenetic hypothesis has been inserted into the Systematic Palaeontology section. Additionally, we take issue with Reviewer 1's strong conviction that his preferred hypothesis for the placement of Megaraptora is correct. Phylogenetic hypotheses are always subject to interpretation based upon consideration of the characters and taxa upon which it is constructed. In our view, the phylogenetic studies upon which Reviewer 1's favoured hypothesis is based (Novas et al. 2013, Porfiri et al. 2014) are insufficient to resolve the affinities of Megaraptora to any degree of certainty. The sampling regime employed included considerably fewer characters than either the most comprehensive basal tetanuran (Carrano et al. 2012) or tyrannosauroid (Carr et al. 2017) datasets. Certain characters used to justify a tyrannosauroid placement of Megaraptora by Porfiri et al. (2014), (e.g., D-shaped premaxillary teeth) have since been shown to be misidentified (Apestiguia et al. 2016). Furthermore, Novas et al. (2016) concluded that the manual anatomy of Australovenator shared many characters with that of Allosaurus, and stated that "Megaraptor and Australovenator are devoid of several manual features that the basal tyrannosauroid Guanlong shares with more derived coelurosaurs (e.g., Deinonychus), thus countering our own previous hypothesis that Megaraptora is well nested within Tyrannosauroidea." Until such time as multiple independent phylogenetic analyses are presented that adequately sample basal tetanuran and tyrannosauroid characters and taxa, and that converge on a single hypothesis for the placement of Megaraptora, we will refer to the diagnosis and placement of the clade (Benson et al. 2010).

Point 5 - We consider that our citation of sources describing variations within megaraptoran taxa and the differing hypotheses for the phylogenetic placement of Megaraptora is presently adequate, as the manuscript under consideration does not make any new contributions or assumptions regarding the morphology of megaraptorans or their affinities.

Points 7, 9, 22, 23, 24, 25 and 26 - The section pertaining to biogeographic implications has been removed.

Point 10, 20 - A section comparing the vertebral and pelvic material with those of the basal ornithomimid *Muttaborrasaurus* and somphospondylan sauropods *Wintonotitan*, *Diamantinasaurus* and *Savannasaurus* has been inserted.

Point 11 - A table containing selected measurements of tetanuran caudal vertebral and pelvic material has been included to provide additional justification for considering the elements LRF 3310-3312 as associated.

Point 12 - Reviewer 1 has misunderstood the reason for the significance of the diagenetic silica on LRF 3310. As has been more clearly phrased in the revised manuscript, the diagenetic silica appears to have preferentially formed around areas of exposed internal bone, which in this case is either broken or eroded surfaces. However, the area in the vicinity of the fossa at the base of the neural arch, which also bears diagenetic silica, does not appear to have been affected by either activity - therefore it is hypothesised that this fossa may have had a pneumatic function. This interpretation is supported by the hypothesised presence of similarly coloured silica "channels" within the neural spine, visible on its broken dorsal surface.

Point 13 - We have acknowledged the possibility of serial variation in the type of vertebral internal structure in the description of LRF 3310-3312.

Point 14, 15 - The pubic peduncle (LRF 3312) is preserved as a pseudomorph, as stated in the Systematic Palaeontology section. We believe that our description adequately compares this specimen with the pubic peduncles of megaraptorans. There is no indication of extensive pneumaticity on the medial or lateral surfaces of the pubic peduncle, as in *Aerosteon* and *Murusraptor*. We do not expect that such obvious features would be erased by the taphonomic processes when the fine ventrolateral striations were left untouched.

Point 16 - Variation in the length to width ratio of the pubic peduncle of the ilium has been recognised as phylogenetically informative for tetanuran theropods. An increase in the relative anteroposterior length of the the pubic peduncle occurred within *Avetheropoda*, with a length to width ratio greater than two diagnosed as a synapomorphy of *Allosauria* (Carrano et al. 2012). Within *coelurosaur*s, at least one described specimen of *Tyrannosaurus* also has pubic peduncles that are considerably longer anteroposteriorly than wide (Brochu 2003), consistent with the aforementioned trend of elongation.

Point 3, 17 - We have rephrased the sentence referred to in Point 17 to imply allosauroid affinities for LRF 3310-3312, with probable carcharodontosaurian affinities under the taxonomic framework we have adopted. We disagree that LRF 3310-3312 is only referable to *Theropoda* indet. and have included, as supplementary information, the results of phylogenetic analyses that include these elements. The presence of vertebral camellae and ventrally keeled proximal caudal vertebrae optimise as synapomorphies of a polytomy including LRF 3310-3312 and carcharodontosaurian taxa in the Carrano et al. (2012) matrix as modified by Apesteguía et al. (2016).

Point 18 - This paragraph has been removed.

Point 19 - This sentence has been removed.

Point 21 - These isolated elements are no longer within the scope the present manuscript and have been removed.

Reviewer 2:

We thank Reviewer 2 for their favourable comments on the quality of the descriptive work presented in the manuscript under consideration. As mentioned previously, we accept the criticism of the palaeobiogeographic discussion and have decided to remove it for resubmission, thus addressing the most serious concerns raised by Reviewer 2.

Appendix B**ROYAL SOCIETY
OPEN SCIENCE****New carcharodontosaurian theropod remains from the mid-Cretaceous Griman Creek Formation, Lightning Ridge (New South Wales, Australia)**

Journal:	Royal Society Open Science
Manuscript ID	RSOS-180826
Article Type:	Research
Date Submitted by the Author:	04-Jun-2018
Complete List of Authors:	Brougham, Tom; University of New England, School of Environmental and Rural Sciences Smith, Elizabeth; Australian Opal Centre Bell, Phil; University of New England, School of Environmental and Rural Sciences
Subject:	Palaeontology < EARTH SCIENCES
Keywords:	Cretaceous, Australia, Dinosauria, Theropoda
Subject Category:	Earth science

Map of Australia showing the location of Lightning Ridge and the mineral claims in which the fossils were
 recovered. The extent of Cretaceous Eromanga and Surat basins in the early to middle Albian is represented
 by the grey area separated by dashed line. The inset map (location indicated by the boxed area) shows the
 area in the vicinity of Lightning Ridge (marked by the star) and the locations of the mineral claims of the
 two theropod occurrences (marked by triangles). Australia coastline uses data taken from GEODATA COAST
 100K 2004 provided by Geoscience Australia (<http://www.ga.gov.au/metadata-gateway/metadata/record/61395>). Basin extents uses data taken from Stewart et al. (2013).

Anterior caudal vertebra LRF 3310 in a, d) posterior, (b,e) right lateral, (c) anterior (f) left lateral (g) ventral views. Boxed area on (b) is expanded in [fig:lrf-3310-fossa]. Abbreviations: aas, anterior articular surface; cc, central convexity; hs, hyposphene; nc, neural canal; ncs, neurocentral suture; nsp, neural spine; pas, posterior articular surface; pcdl, posterior centrodiapophyseal lamina; poz, postzygapophysis. Scale bar equals 50 mm.

Anterior caudal vertebra LRF LRF 3311 in (a) posterior, (b) anterior, (c,d) lateral and (e) ventral views. Abbreviations: cf, chevron articular facet; nc, neural canal. Scale bar equals 50 mm.

Right pubic peduncle of the ilium LRF 3312 in (a) lateral, (b) medial, (c) dorsal and (d) ventral. Scale bar equals 50 mm.

Vertebral articular end AM F106525. a) articular surface of the centrum; b) view of broken surface of the centrum; c) ventral surface; d, e) lateral surfaces; f) oblique dorsal view; g) interpretive drawing of the exposed internal structure of the centrum, grey indicates presence of interior septa overlying the camerae. Abbreviations: cc, central convexity; ca, camerae; se, septa. Scale bar equals 20 mm.

Vertebral centrum AM F112816 in a) right lateral, b) anterior, c) dorsal, d) left lateral, e) posterior, and f) ventral views. Abbreviations: ca, camellae; cf, chevron facet; pf, pneumatic foramina; vk, ventral keel. Scale bar equals 20 mm.

Measurement	LRF 3310	LRF 3311	AM F112816
Centrum, anterior articular surface, width	47.3 -		?
Centrum, anterior articular surface, height	58.5 -		27
Centrum, posterior articular surface, width	101.6	73.4 ?	
Centrum, posterior articular surface, height	93.3 -		32.4
Centrum, anteroposterior length	58.3	65.6	44.9
Centrum, mediolateral width at mid-length	41.5	26.7	17.1
Centrum, dorsoventral height at mid-length	58.1 -		?
Neural arch, height		20.3 -	-
Neural canal, width	30.8*	-	-
Neural canal, height	20.3*	-	-

Taxon	Source	Ilium pubic peduncle			
		Length	Width	Length ÷ height	
LRF 3310/3313	-		152	78	1.94
Aerosteon	Sereno et al. (2008)		169	81	2.09
Ichthyovenator	Allain et al. (2012)		138	75	1.84
Majungasaurus	O'Connor (2007); Carrano (2007)	82*	62*	1.32*	
Neovenator	Brusatte et al. (2008)		135	67	2.01

Caudal vertebra 1

Posterior width	Posterior height	Ilium pubic peduncle ÷ caudal vertebra 1 width
101.6	93	0.77
128	118	0.63
141	120	0.53
52.3	58.8	1.19
100*	114	0.67

New carcharodontosaurian theropod remains from the mid-Cretaceous Griman Creek Formation, Lightning Ridge (New South Wales, Australia)

Tom Brougham^{1*}, Phil R. Bell¹, Elizabeth T. Smith²

¹School of Environmental and Rural Science, University of New England, Armidale 2351, NSW, Australia

²Australian Opal Centre, 3/11 Morilla Street, Lightning Ridge 2834, NSW, Australia

*tbrougha@myune.edu.au

Abstract

The limited fossil record of Australian Cretaceous theropods is dominated by megaraptorans, reported from associated and isolated material from the Early Cretaceous of Victoria and the 'mid'-Cretaceous of central-north New South Wales and central Queensland. Here, we report on new postcranial theropod material from the early Late Cretaceous Griman Creek Formation at Lightning Ridge. Among this new material is an associated set consisting of two anterior caudal vertebrae and a pubic peduncle of the ilium. These elements display a combination of characteristics typically associated with carcharodontosaurian theropods, include camellate internal composition of the vertebral centra, ventrally keeled anterior caudal centra and a pubic peduncle of the ilium with a ventral surface approximately twice as long anteroposteriorly as mediolaterally wide. The absence of pneumaticity in the pubic peduncle and the anterior caudal centra contrasts with its presence in all megaraptorids in which those elements are preserved, indicating that megaraptorid affinities for this material are unlikely. A morphologically similar partial vertebral centra also from the Griman Creek Formation is tentatively referred with this material, which differs in bearing a camerate internal composition. This new material is distinct from previously-described Lightning Ridge megaraptorid material and thus represents a second carcharodontosaurian from this interval. Additionally, a mid-caudal vertebral centrum bearing pneumatic foraminae and extensive

camellae is referred to Megaraptora and is the first axial skeletal element of a megaraptorid
allosauroid described from Lightning Ridge.

8 **Introduction**

[revised manuscript text omitted]

**Systematic framework**

In the proceeding descriptions and discussion, Megaraptoridae is considered to be nested
within Neovenatoridae, which together with its sister clade Carcharodontosauridae forms
Carcharodontosauria [48]. Recently proposed placements of Megaraptoridae within
Tyrannosauroidae [10,49,50] or as the sister taxon of Coelurosauria [51] are acknowledged as
alternative hypotheses. However, as noted elsewhere [52], only one phylogenetic dataset so
far supports these novel relationships; corroboration by independent phylogenetic analyses
that incorporate a broader sampling of basal tetanuran and basal coelurosaurian characters
and taxa is required before either hypothesis can be accepted over the current consensus
view. Nomenclature for description of vertebral laminae and fossae follows that of [53] and
[54] respectively.

**Systematic palaeontology**

Dinosauria Owen, 1842

Theropoda Marsh, 1881

Tetanurae Gauthier, 1986

Allosauroidae Currie and Zhao, 1993

Carcharodontosauria Benson, Carrano and Brusatte, 2010

Material

Two anterior (LRF 3310, LRF 3311) and one (?)mid (AM F112816) caudal vertebrae, a right pubic peduncle of the ilium (LRF 3312), and a centrum articular surface (AM F106525). LRF 3310–3312 were recovered from a one metre diameter drill shaft in the eastern section of Smiths Field on the Coocoran opal field, approximately 20 km west of Lightning Ridge (Fig. 1). Their close association within the thin opal- and fossil-bearing layer of the GCF and the absence of overlapping material or other taxa in the immediate vicinity indicates that they pertain to a single individual. AM F106525 was recovered from a mineral claim known as ‘The Bone Yard’ at the Nine Mile field, approximately 8 km west-northwest of Lightning Ridge (Fig. 1).

Preservation

LRF 3310–3312

LRF 3310 represents an almost complete centrum, the posterior part of the neural arch, and the base of the right transverse process (Fig. 2). The anterior end of the vertebral centrum has been abraded but is intact (Fig. 2b). The posterior articular surface of the centrum is well preserved but is missing a portion of the left rim (Fig. 2a). The left lateral surface of the centrum has been crushed (Fig. 2d), resulting in a rightward displacement of the anterior end of the centrum in ventral view (Fig. 2e). Diagenetic veins of silica have formed in and around the crushed area on the left side of the centrum; this mineralisation can also be seen on the neural spine and postzygapophysis (Fig. 2d, Fig. 3). Of the neural arch, only the right postzygapophysis and the bases of the right transverse process and neural spine are preserved. The edges of the articular surface of the postzygapophysis have been eroded (Fig. 2c).

LRF 3311 represents the ventral portion of a centrum and a dorsal fragment of the posterior articular surface (Fig. 4). The anterior articular end of the centrum has been broken off, exposing an internal cavity (Fig. 4b); no internal structures can be discerned along the plane of the break.

LRF 3312 is interpreted as representing the ventral end of the pubic peduncle of a right ilium. The broken and exposed dorsal surface is mediolaterally thin, and the interior of the bone

appears to have been preserved as a solid mass of opal, obscuring any detail of the original
bone texture. The lateral surface is well preserved whereas only the dorsalmost portion of the
medial surface of the peduncle is visible through the adherent matrix (Fig. 5a,b). The ventral
surface, where it would have contacted the proximal pubis, is heavily eroded and densely
covered in matrix. On the concave posteroventral surface, two subcircular depressions are
present (Fig. 5d) that are inferred to be possible bioerosional features, and as such do not
represent an original feature of the bone. Only the ventrolateral portion of the acetabular
margin is preserved.

**AM F106525**

AM F106525 are both isolated articular end of a centrum. In AM F106524, only a small

[revised manuscript text omitted]

on the anterior caudals of *Alioramus altai* may imply ontogenetic or individual variation,
consistent with apneumatic functions such as the sites of axial musculature attachments or fat
deposits [70,93].
The subtriangular pubic peduncle of the ilium is similar in appearance to that of known ilia
from megaraptoran allosauroids. The lateral surface bearing fine parallel striations is
interpreted as the attachment site of connective tissue between the pubic peduncle and the
pubis; such striations with this inferred function have also been observed on the pubic
peduncles of megaraptorans [19,60]. The peduncle appears to be solid with no indication of
pneumaticity visible on the lateral surface or in cross-section on the broken dorsal surface.
This contrasts with the condition of neovenatorid allosauroids (including megaraptorans) in
which pneumatic chambers penetrate the ilia through the medial surface, the brevis fossa,
and/or the pubic peduncle [19,48,60]. In LRF 3312, the anteroposterior length of the ventral
surface of the pubic peduncle is approximately twice the mediolateral width, similar to that
reported for carcharodontosaurian allosauroids, and which is intermediate between the
relatively shorter pubic peduncles of non-allosaurian tetanurans and the longer ones of
coelurosaurs Table 2, [18].

On the basis of the characters discussed above, LRF 3310–3312 presents a combination of
characters that are commonly observed in carcharodontosaurian allosauroids: camellae in the
caudal centra, a ventral keel on the anterior caudal centra and a pubic peduncle
approximately twice as long anteroposteriorly as mediolaterally wide [48]. This hypothesis is
supported by the inclusion of LRF 3310–3312 into a phylogenetic dataset which resulted in
the former two characters representing unambiguous synapomorphies of a polytomy
containing exclusively carcharodontosaurian allosauroids (Supplementary Figure 1); inclusion
in a secondary matrix with a less inclusive sampling of characters and taxa hypothesises more
general allosauroid affinities (Supplementary Figure 2). In addition, LRF 3310 and AM
F106525 both have in common a central convexity on the articular surface of the centrum, an
uncommon feature among theropods. The shared presence of this unusual characteristic in
both vertebrae, together with their relative proximity to each other suggests that they may
pertain to the same taxon, or similar taxa, of carcharodontosaurian allosauroid. The
differences in internal structure of LRF 3310 and AM F106525 would not preclude the

possibility of a close relationship as there is well-documented variation in vertebral internal
composition within individual carcharodontosaurian taxa.

Among Australian megaraptorids, the pubic peduncle of LRF 100–106 bears a pneumatic
internal composition; the fragment of the main body of the ilium in *Australovenator* shows
evidence of pneumaticity [11], but it is not known if the pubic peduncle was pneumatic. As
there is no indication of pneumaticity in LRF 3312, it is therefore distinguishable from LRF
100–106. The absence of any vertebral material from either LRF 100–106 or *Australovenator*
unfortunately limits the extent to which any additional direct comparisons between the three
taxa can be made. However, the pervasive development of both pelvic and caudal
pneumaticity among megaraptorids [60] indicates that LRF 3310–3312 most likely does not
pertain to this theropod clade.
AM F112816 most likely pertains to a megaraptoran, primarily on the basis of pneumaticity
within a mid-caudal centra. The only other theropod clade in which pneumatic mid-caudal
vertebrae have been reported, Oviraptorosauria, is unlikely to be a candidate for the affinities
of this centrum as the group is likely to have been entirely absent from Gondwana. Restudy of
previous material referred to Oviraptorosauria from South America [97,98] has concluded
that they are representative of noosaurids and megaraptorans respectively [99,100].

To date, a majority of the material of Australian apex theropod predators has been referred to
Megaraptora [2,5,19,48]. Recently, the Victorian Otway and Gippsland groups have yielded
individual specimens interpreted as representatives of other medium and large-sized
theropod clades, including a tyrannosauroid pubis, a spinosaurid cervical vertebra, and a
ceratosaurian astragalocalcaneum [6–8]. The diagnoses for each of these remains were
subsequently disputed, with the tyrannosauroid pubis interpreted as either an indeterminate
tetanuran or possible megaraptoran, [9,10], and the spinosaurid and ceratosaurian specimens
as indeterminate averostrans [10]. While a re-evaluation of the problematic Victorian
material is beyond the scope of this present study, it is noted that if the interpretation of LRF
3310–3312 is correct, then this specimen represents additional evidence for a non-
megaraptoran Australian Cretaceous theropod, and the only one that has been described from
associated skeletal elements.

Conclusion

New associated and isolated material from the Upper Cretaceous Griman Creek Formation expands the diversity of theropods currently recognised from this stratigraphic interval. LRF 3310–3312, tentatively grouped together with AM F106524–106525, represents a second medium-sized carcharodontosaurian theropod from Lightning Ridge, distinct from the megaraptorid LRF 100–106, and is only the third Australian theropod recognised from associated material. AM F112816 is identified as a megaraptorid mid-caudal vertebra due to the presence of pneumatic foraminae and camellate internal structure, and is the first reported axial skeletal element of such a theropod recognised from Lightning Ridge.

Data Accessibility

Supplementary information has been provided as the electronic supplementary material.

Authors' Contributions

T.B. and P.R.B. conceived the study; T.B. wrote the manuscript; P.R.B. and E.S. helped draft the manuscript; T.B. and E.S. collected measurements; T.B. performed analyses. All authors gave final approval for publication.

Competing Interests

We declare we have no competing interests.

Funding

TB was funded by a Research Training Program scholarship. PRB was funded under an Australian Research Council Discovery Early Career Researcher Award (project ID: DE170101325).

Research Ethics

No ethics approval was required prior to conducting this research.

Animal Ethics

No ethics approval was required prior to conducting this research.

Permission to carry out fieldwork

The material described in this research was deposited in museum collections. No fieldwork was undertaken prior to conducting this research.

Acknowledgements

LRF 3300–3312 were gifted to the Australian Opal Centre by one of the authors (ES). Jenni Brammall, Manager of the Australian Opal Centre, is also thanked for allowing access to the specimens and providing resources to facilitate their study while in Lightning Ridge. Matthew McCurry of the Australian Museum is thanked for allowing access to material under his care. TNT is made available thanks to a subsidy from the Willi Hennig Society. We acknowledge the Yuwaalaraay, and Gamilaraay peoples, traditional owners of Lightning Ridge country.

[revised manuscript text omitted]

52. Brusatte SL, Carr TD. 2016 The phylogeny and evolutionary history of tyrannosauroid
dinosaurs. *Scientific Reports* **6**, 20252. (doi:[10.1038/srep20252](https://doi.org/10.1038/srep20252))
53. Wilson JA. 1999 A nomenclature for vertebral laminae in sauropods and other saurischian
dinosaurs. *Journal of Vertebrate Paleontology* **19**, 639–653.
(doi:[10.1080/02724634.1999.10011178](https://doi.org/10.1080/02724634.1999.10011178))
54. Wilson JA, D’Emic MD, Ikejiri T, Moacdieh EM, Whitlock JA. 2011 A nomenclature for
vertebral fossae in sauropods and other saurischian dinosaurs. *PLoS ONE* **6**, e17114.
(doi:[10.1371/journal.pone.0017114](https://doi.org/10.1371/journal.pone.0017114))
55. Currie PJ, Zhao X-J. 1993 A new carnosaur (Dinosauria, Theropoda) from the Jurassic of
Xinjiang, People’s Republic of China. *Can. J. Earth Sci.* **30**, 2037–2081. (doi:[10.1139/e93-179](https://doi.org/10.1139/e93-179))
56. Bartholomai A, Molnar RE. 1981 Muttaborrasaurus, a new iguanodontid (Ornithischia:
Ornithopoda) dinosaur from the Lower Cretaceous of Queensland. *Memoirs of the Queensland*
*Museum* **20**, 319–349.
57. Poropat SF, Mannion PD, Upchurch P, Hocknull SA, Kear BP, Elliott DA. 2015 Reassessment
of the non-titanosaurian somphospondylan *Wintonotitan Wattsi* (Dinosauria: Sauropoda:
Titanosauriformes) from the mid-Cretaceous Winton Formation, Queensland, Australia.
*Papers in Palaeontology* **1**, 59–106. (doi:[10.1002/spp2.1004](https://doi.org/10.1002/spp2.1004))
58. Poropat SF *et al.* 2016 New Australian sauropods shed light on Cretaceous dinosaur
palaeobiogeography. *Scientific Reports* **6**, 34467. (doi:[10.1038/srep34467](https://doi.org/10.1038/srep34467))
59. Poropat SF, Upchurch P, Mannion PD, Hocknull SA, Kear BP, Sloan T, Sinapius GHK, Elliott
DA. 2015 Revision of the sauropod dinosaur *Diamantinasaurus Matildae* Hocknull *et al.* 2009
from the mid-Cretaceous of Australia: Implications for Gondwanan titanosauriform dispersal.
*Gondwana Research* **27**, 995–1033. (doi:[10.1016/j.gr.2014.03.014](https://doi.org/10.1016/j.gr.2014.03.014))
60. Sereno PC, Martinez RN, Wilson JA, Varricchio DJ, Alcober OA, Larsson HCE. 2008 Evidence
for avian intrathoracic air sacs in a new predatory dinosaur from Argentina. *PLoS ONE* **3**,
e3303. (doi:[10.1371/journal.pone.0003303](https://doi.org/10.1371/journal.pone.0003303))

61. Allain R, Xaisanavong T, Richir P, Khentavong B. 2012 The first definitive Asian
spinosaurid (Dinosauria: Theropoda) from the early cretaceous of Laos. *Naturwissenschaften*
**99**, 369–377. (doi:[10.1007/s00114-012-0911-7](https://doi.org/10.1007/s00114-012-0911-7))
62. O'Connor PM. 2007 The postcranial axial skeleton of *Majungasaurus Crenatissimus*
(Theropoda: Abelisauridae) from the Late Cretaceous of Madagascar. *Journal of Vertebrate*
*Paleontology* **27**, 127–163.
63. Carrano MT. 2007 The appendicular skeleton of *Majungasaurus Crenatissimus*
(Theropoda: Abelisauridae) from the Late Cretaceous of Madagascar. *Journal of Vertebrate*
*Paleontology* **27**, 163–179.
64. Brusatte SL, Benson RBJ, Hutt S. 2008 The osteology of *Neovenator Salerii* (Dinosauria:
Theropoda) from the Wealden Group (Barremian) of the Isle of Wight. *Monograph of the*
*Palaeontographical Society Series* **162**, 1–75.
65. Allain R, Chure DJ. 2002 *Poekilopleuron bucklandii*, the theropod dinosaur from the Middle
Jurassic (Bathonian) of Normandy. *Palaeontology* **45**, 1107–1121. (doi:[10.1111/1475-](https://doi.org/10.1111/1475-4983.00277)
[4983.00277](https://doi.org/10.1111/1475-4983.00277))
66. Méndez AH. 2012 The caudal vertebral series in abelisaurid dinosaurs. *Acta*
*Palaeontologica Polonica* **59**, 99–107. (doi:[10.4202/app.2012.0095](https://doi.org/10.4202/app.2012.0095))
67. Madsen SK. 1976 *Allosaurus fragilis*: A revised osteology. *Utah Geological Survey Bulletin*
**109**, 1–163.
68. Sadleir RW, Barrett PM, Powell HP. 2008 The anatomy and systematics of
*Eustreptospondylus Oxoniensis*. *Monograph of the Palaeontographical Society, London* **160**, 1–
82.
69. Benson RB. 2008 New information on *Stokesosaurus*, a tyrannosauroid (Dinosauria:
Theropoda) from North America and the United Kingdom. *Journal of vertebrate Paleontology*
**28**, 732–750.

[revised manuscript text omitted]

88. Coria RA, Currie PJ. 2016 A new megaraptoran dinosaur (Dinosauria, Theropoda,
Megaraptoridae) from the Late Cretaceous of Patagonia. *PLOS ONE* **11**, e0157973.
(doi:[10.1371/journal.pone.0157973](https://doi.org/10.1371/journal.pone.0157973))
89. Britt BB. 1991 Theropods of Dry Mesa Quarry (Morrison Formation, Late Jurassic),
Colorado, with emphasis on the osteology of *Torvosaurus Tanneri*. *Brigham Young University*
*Geology Studies* **37**, 1–72.
90. Xu X, Tan Q, Wang J, Zhao X, Tan L. 2007 A gigantic bird-like dinosaur from the Late
Cretaceous of China. *Nature* **447**, 844–847. (doi:[10.1038/nature05849](https://doi.org/10.1038/nature05849))
91. Balanoff AM, Norell MA. 2012 Osteology of *Khaan Mckennai* (Oviraptorosauria:
Theropoda). *Bulletin of the American Museum of Natural History* **372**, 1–77.
(doi:[10.1206/803.1](https://doi.org/10.1206/803.1))
92. Benson RBJ, Butler RJ, Carrano MT, O'Connor PM. 2012 Air-filled postcranial bones in
theropod dinosaurs: Physiological implications and the 'reptile'–bird transition. *Biological*
*Reviews* **87**, 168–193. (doi:[10.1111/j.1469-185X.2011.00190.x](https://doi.org/10.1111/j.1469-185X.2011.00190.x))
93. O'Connor PM. 2006 Postcranial pneumaticity: An evaluation of soft-tissue influences on
the postcranial skeleton and the reconstruction of pulmonary anatomy in archosaurs. *Journal*
*of Morphology* **267**, 1199–1226. (doi:[10.1002/jmor.10470](https://doi.org/10.1002/jmor.10470))
94. Zhao X-J, Currie PJ. 1993 A large crested theropod from the Jurassic of Xinjiang, People's
Republic of China. *Can. J. Earth Sci.* **30**, 2027–2036. (doi:[10.1139/e93-178](https://doi.org/10.1139/e93-178))
95. Kobayashi Y, Barsbold R. 2005 Anatomy of *Harpymimus Okladnikov* Barsbold and Perle
1984 (Dinosauria; Theropoda) of Mongolia. In *The Carnivorous Dinosaurs* (ed K Carpenter),
pp. 97–126. Bloomington: Indiana University Press.
96. Zhao X, Benson RBJ, Brusatte SL, Currie PJ. 2010 The postcranial skeleton of
*Monolophosaurus Jiangi* (Dinosauria: Theropoda) from the Middle Jurassic of Xinjiang, China,
and a review of Middle Jurassic Chinese theropods. *Geological Magazine* **147**, 13–27.
(doi:[10.1017/S0016756809990240](https://doi.org/10.1017/S0016756809990240))

97. Frey E, Martill DM. 1995 A possible oviraptorosaurid theropod from the Santana
Formation (Lower Cretaceous,? Albian) of Brazil. *Neues Jahrbuch fur Geologie und*
*Palaontologie Monatshefte* **7**, 397–412.
98. Frankfurt NG, Chiappe LM. 1999 A possible oviraptorosaur from the Late Cretaceous of
northwestern Argentina. *Journal of Vertebrate Paleontology* **19**, 101–105.
(doi:[10.1080/02724634.1999.10011126](https://doi.org/10.1080/02724634.1999.10011126))
99. Aranciaga Rolando AM, Egli FB, Sales MAF, Martinelli AG, Canale JI, Ezcurra MD. 2018 A
supposed Gondwanan oviraptorosaur from the Albian of Brazil represents the oldest South
American megaraptoran. *Cretaceous Research* **84**, 107–119.
(doi:[10.1016/j.cretres.2017.10.019](https://doi.org/10.1016/j.cretres.2017.10.019))
100. Agnolin FL, Martinelli AG. 2007 Did oviraptorosaurs (Dinosauria; Theropoda) inhabit
Argentina? *Cretaceous Research* **28**, 785–790. (doi:[10.1016/j.cretres.2006.10.006](https://doi.org/10.1016/j.cretres.2006.10.006))

Appendix C

Note to the Editor:

In considering present and future comments regarding this manuscript, we would like to draw the Editor's attention to the uncharacteristically strong (in our opinion) defence Reviewer 1 has consistently given to his preferred phylogenetic hypotheses, the subject of which was never a key component of the research presented in this work. In addition, we would also like to remind and emphasise to the Editor that the two other reviewers who have provided comment on this manuscript at various stages have taken comparatively little issue with the use of any particular taxonomic framework, emphasising that the views of Reviewer 1 are not held by all researchers engaged in theropod taxonomy. As is detailed in the response below, we believe we have dealt with the ultimate cause of Reviewer 1's primary concerns by considering phylogenetic debates surrounding Megaraptora as unresolved, and using terms acceptable and interpretable under any given hypothesis. In the event that further issues are raised with respect to phylogenetic hypotheses as discussed in the present manuscript, we hope that the Editor will consider them in the context of an ongoing and unresolved problem within theropod phylogeny and the aforementioned neutral framing of our taxonomic discussion.

In this revision of the manuscript we have taken the opportunity to include an additional specimen of a theropod from Lightning Ridge that has recently come to our attention. This supplements the material already described and does not alter our conclusions.

Response to reviewers comments

Reviewer 1:

1. As will be discussed in more detail below, the referral of LRF 3310–3312 has been modified to the more inclusive theropod clade Avetheropoda, which reflects certain ambiguities in the combination and number of identifiable characters as preserved. We believe that this obviates a statement in this comment that this manuscript makes novel claims about Australian Cretaceous theropod faunal composition. In addition, we more clearly outline our position with respect to the debate surrounding megaraptoran phylogenetic hypotheses, opting to use definitions that can be interpreted independent of any specific topology. As such, the manuscript now

no longer makes any additional claims, or offers any support for or against, any of the current phylogenetic, and by extension palaeobiogeographic, hypotheses pertaining to Megaraptora. Instead, the principal purpose of this manuscript is to illustrate, describe and discuss in a limited fashion new material pertaining to theropod dinosaurs, a group which is still poorly represented in Australia.

2. The phrasing of this sentence has been modified to reflect the predominance of abelisaurids in Cretaceous Patagonia and the proportion of carcharodontosaurids.
3. We have removed references to the terms “Carcharodontosauria” and “carcharodontosaurian(s)” that are specific to the phylogenetic hypothesis of Benson et al. (2010) in line with the suggestion offered. However, their removal does not imply tacit acceptance of either of the alternative hypotheses suggested by Reviewer 1, but instead reflects the present uncertainty of the phylogenetic placement of Megaraptora. In my revised Systematic Framework section, we briefly outline the debate surrounding various competing hypotheses of megaraptoran affinities and state my preference for using the composition of Megaraptora and Megaraptoridae as defined by Novas et al. (2013), which has remained relatively stable between all three current hypotheses, but not necessarily their preferred phylogenetic hypothesis. we consider this to be a conservative approach in the present situation, and is an attempt to ensure that the descriptions and discussions presented in this manuscript can remain relevant if and when the aforementioned controversies are resolved.
4. The sentence in the Systematic Framework section that prompted this statement has been removed and replaced with the aforementioned discussion of the various competing hypotheses for the placement of Megaraptora and our decision to use taxonomic descriptors that are ambivalent to any of the presently offered hypotheses. We strongly note that the present manuscript does not in any way attempt to make any substantial contribution to the debate as it stands; however, we will take this opportunity to respond to this comment. Reviewer 1 is correct that the Novas et al. (2013) dataset was more inclusive in its taxonomic scope than that of Benson et al. (2010). However, our principal concern with the Novas et al. (2013) dataset, as stated in my previous response, is that its sampling is considerably reduced in comparison to the two source datasets Reviewer 1 claims were unified in its construction (i.e., 61 unique genera in Benson et al. [2010] and Brusatte et al. [2010]

combined, as opposed to 44 unique genera in Novas et al. [2010]; 540 total characters Benson et al. [2010] and Brusatte et al. [2010] combined, with few shared characters, as opposed to 287 characters from Novas et al. [2013]). Consideration of all available evidence is of the utmost importance when assessing the robustness of any given phylogenetic hypothesis; this has in fact been demonstrated in the revised version of the Novas et al. (2013) dataset by Apesteguia et al. (2016), in which the addition of taxa and characters hypothesised a phylogenetic position of Megaraptora as the sister taxon of Coelurosauria, as opposed to Reviewer 1's favoured position within Tyrannosauroidea. In light of the present uncertainties as discussed above and in the revised manuscript, we believe that the strength of Reviewer 1's defence of his preferred hypothesis is premature and misplaced. Nonetheless, Novas et al. (2013) and Apesteguia et al. (2016) represent important contributions to the ongoing debate surrounding megaraptoran phylogenetics, but we maintain that an increase in both the sampling of characters and taxa presented in both datasets. that forms the basis for the phylogenetic hypotheses offered, is ultimately required if resolution is to be achieved.

5. We have relaxed our diagnosis of LRF 3310–3312 to the more inclusive clade Avetheropoda, which encompasses all the taxa to which Reviewer 1 has drawn comparisons in his comments. However, contrary to the position outlined in this comment, it would be improper to refer this material to Megaraptoridae as there are no recognisable autapomorphies present to support such a claim. The fact that megaraptorids thus far comprise a large percentage of the Australian Cretaceous theropod fauna is insufficient grounds for claiming that any theropod material found in Australia during this time interval must by default pertain to a megaraptorid.
6. Statements to this effect were inserted in the previous revision; these have now been augmented with passages mentioning the phylogenetic ambiguity involved in interpreting patterns of vertebral pneumatic composition in theropods.
7. Theses observation has been added to the discussions where appropriate.
8. The taxa listed here all fall within the theropod clade Avetheropoda, to which LRF 3310–3312 has now been assigned, albeit indeterminately.

9. and 10. See previous comments about the relaxing of the taxonomic status of LRF 3310–3312 to an indeterminate avetheropodan.

Reviewer 2.

We thank reviewer 2 for his favourable comments to the manuscript under consideration. We have addressed the concerns outlined in his review and present them in this revised version.

1. As stated above, the Systematic Framework section has been revised, and potentially contentious passages have been removed and replaced with a discussion more clearly outlining the rationale for our hierarchical taxonomy. In addition, clades specific to particular phylogenetic hypotheses of Megaraptora (i.e., Neovenatoridae, Carcharodontosauridae) have been replaced with suitable alternatives.
2. We thank Reviewer 2 for bringing our attention to the extent of development of pneumaticity in the caudal vertebrae of *Aoniraptor*; this information has been incorporated into our discussion of LRF 3310, LRF 3311 and AM F112816.